# High-Dimensional Learning Dynamics of Quantized Models with Straight-Through Estimator

**Yuma Ichikawa** [1] [2]  **Shuhei Kashiwamura** [3] [4]  **Ayaka Sakata** [2] [5]

## Abstract

Quantized neural network training optimizes a discrete, non-differentiable objective. The straight-through estimator (STE) enables backpropagation through surrogate gradients and is widely used. While previous studies have primarily focused on the properties of surrogate gradients and their convergence, the influence of quantization hyperparameters, such as bit width and quantization range, on learning dynamics remains largely unexplored. We theoretically show that in the high-dimensional limit, STE dynamics converge to a deterministic ordinary differential equation. This reveals that STE training exhibits a plateau followed by a sharp drop in generalization error, with plateau length depending on the quantization range. A fixed-point analysis quantifies the asymptotic deviation from the unquantized linear model. We also extend analytical techniques for stochastic gradient descent to nonlinear transformations of weights and inputs.

## 1. Introduction

Deep neural networks (DNNs) have transformed machine learning, with remarkable success in computer vision (He et al., 2016; Krizhevsky et al., 2012), reinforcement learning (Mnih et al., 2013; Sutton et al., 1999), and natural language processing (Achiam et al., 2023; Touvron et al., 2023; Grattafiori et al., 2024). However, state-of-the-art DNNs often contain billions of parameters, making inference costly in terms of compute, memory, and energy. These requirements prevent deployment on resource-constrained devices such as smartphones and Internet of Things (IoT) devices (Liang et al., 2021).

Quantization has emerged as a widely used approach. By representing weights and activations with a few bits, quantized DNNs replace full-precision floating-point operations with low-precision arithmetic, substantially reducing the memory footprint and computation while preserving accuracy. Training quantized DNNs, or quantization-aware training (QAT) (Cai et al., 2017; Choi et al., 2018; Hubara et al., 2018; Rastegari et al., 2016; Yin et al., 2018), is challenging because quantization makes the loss function piecewise constant in the quantized variables, thereby preventing standard backpropagation. A widely adopted method is the straight-through estimator (STE) (Bengio et al., 2013), which uses a surrogate gradient in the backward pass. Despite being a heuristic modification of the chain rule, this approach has enabled low-bit training across a range of tasks and architectures. Beyond quantization, STE has become a general tool for discrete objectives, including neural architecture search (Riad et al., 2022; Stamoulis et al., 2020), discrete latent variables (Jang et al., 2016; Kunes et al., 2023; Paulus et al., 2020), adversarial attacks (Athalye et al., 2018), and sparse recovery (Mohamed et al., 2023), among others (Mao et al., 2022; Wagstaff et al., 2022; Xu & Li, 2019).

Despite its empirical success, STE lacks theoretical justification, raising concerns about the stability and performance of the optimization process. Efforts to clarify the properties of STE have progressed in recent years. Recent analyses have clarified when STE-based updates serve as descent directions and when design choices can destabilize learning (Li et al., 2017; Yin et al., 2019a). However, the influence of quantization hyperparameters, such as bit width and quantization range, on learning dynamics remains underexplored. Complementary work has examined generalization error as a function of quantization hyperparameters without assuming a specific training rule (Kashiwamura et al., 2024). However, these results cannot capture the dynamics induced by the STE. Moreover, most existing studies address either weight-only quantization (Kashiwamura et al., 2024; Ajanthan et al., 2021; Dockhorn et al., 2021; Jin et al., 2025; Liu et al., 2023b) or activation-only quantization (Long et al., 2021; Yin et al., 2019a), whereas jointly quantizing weights and inputs is more challenging and has recently attracted significant attention in DNN quantization.

We study a minimal yet representative linear regression

[1]Fujitsu Limited [2]Riken Center for AIP [3]The University of Tokyo [4]NTT Research, Inc. [5]Ochanomizu University. Correspondence to: Yuma Ichikawa <ichikawa.yuma@fujitsu.com>.

*Proceedings of the 43rd International Conference on Machine Learning*, Seoul, South Korea. PMLR 306, 2026. Copyright 2026 by the author(s).

model with jointly *quantized weights and inputs*, trained using STE in the high-dimensional input limit, which is closely related to layer-wise post-training quantization (PTQ) (Frantar et al., 2022; Lin et al., 2024; Arai & Ichikawa, 2025). This analysis enables us to quantify how STE dynamics depend on quantization hyperparameters and to characterize phenomena such as non-monotonic behavior at low bit widths, stability improvements due to quantization, and the performance gap relative to the unquantized model.

**Contribution** We develop a theoretical framework that characterizes the learning dynamics of STEs. Our contributions are as follows.

- In the high-dimensional limit, the microscopic parameter updates converge to a continuous-time stochastic differential equation (SDE), whereas the macroscopic states follow a deterministic ordinary differential equation (ODE). We also extend the high-dimensional dynamics analyses to settings with nonlinear transformations of both parameters and inputs.

- The ODEs predict a typical two-phase trajectory: an extended plateau followed by a sharp drop in generalization error and eventual saturation. The timing depends on the quantization range, underscoring the importance of carefully selecting hyperparameters.

- Input quantization can significantly degrade performance. Furthermore, in the low-bit regime, the dynamics of the STE become non-monotonic, leading to slower convergence.

- Asymptotic stability analysis identifies the regimes where quantization preserves training stability, even at higher learning rates, thereby clarifying when quantization acts as an implicit regularizer rather than a mere perturbation. We also quantify the performance degradation relative to the unquantized model and express it as an explicit function of the quantization hyperparameters.

**Reproducibility** The code to reproduce our results is available in the supplementary material.

## 2. Related Work

**STE Dynamics.** Research on the theoretical underpinnings of STE has focused primarily on activation quantization in two-layer neural networks. It has been shown that with appropriate surrogates, STE reduces population risk in regression and, under appropriate conditions, can converge to a perfectly separating classifier, whereas poor surrogates may induce instability (Yin et al., 2019a;b; Long et al., 2021). Furthermore, clipped STE has been shown to

break scale invariance and thereby promote convergence to empirical risk minima (Gongyo & Ishikawa, 2022). Beyond these settings, STE has also been formulated as a Wasserstein gradient flow to study convergence properties (Cheng et al., 2019), and interpreted as a first-order approximation (Liu et al., 2023a). To the best of our knowledge, however, the STE dynamics in jointly weight-input quantized models remain unexplored.

**Static Properties of Quantization.** The static properties of quantized models are increasingly understood. In terms of representation ability, theoretical results characterize how approximation power scales with bit-width and the parameter overhead required to match full-precision networks (Ding et al., 2018), and also establish the bit precision necessary to achieve unquantized approximation rates (Gonon et al., 2023). With respect to generalization, PAC-Bayes and description-length analyses suggest that low-bit models have tighter generalization bounds (Lotfi et al., 2022). Some studies interpret quantization as an implicit regularizer that biases optimization toward flatter minima, often improving generalization (AskariHemmat et al., 2024; Javed et al., 2024). Statistical physics analyses indicate that, in simple classification settings, performance saturates beyond a small bit width (Baldassi et al., 2016). In addition to bit-width, recent work (Kashiwamura et al., 2024) reveals how the quantization range affects generalization and the double descent phenomenon. Despite progress in static analysis, the training dynamics of quantized models remain largely unexplored, particularly in terms of how bit width and range influence learning.

**Learning Dynamics in High-dimensional Limit.** Deterministic dynamical descriptions of SGD in the high-dimensional input limit have been studied within the statistical physics community. This line of studies began with single- and two-layer neural networks featuring a few hidden units (Kinzel & Rujan, 1990; Kinouchi & Caticha, 1992; Copelli & Caticha, 1995; Biehl & Schwarze, 1995; Riegler & Biehl, 1995; Vicente et al., 1998; Yoshida & Okada, 2019), based on a heuristic derivation of ODEs that describe typical learning dynamics. These results have recently been rigorously proven using the concentration phenomena in stochastic processes (Ichikawa & Hukushima, 2024; Wang et al., 2018; 2019; Goldt et al., 2019; Veiga et al., 2022). The key idea in this type of analysis is to capture high-dimensional parameter dynamics with a few macroscopic variables. However, this analytical method is restricted to systems without nonlinear transformation of both parameters and inputs. Therefore, it is difficult to directly apply this method to the analysis of STE dynamics that include nonlinear transformations of both weights and inputs.

## 3. Problem Setup

**Notation.** We summarize the notation used in this study. For $T \in \mathbb{N}$, we define $[T] := \{1, \ldots, T\}$. Vectors are denoted by bold lowercase letters, e.g., $\boldsymbol{x}$, and matrices by upper-case letters, e.g., $X$. The standard normal density and distribution function are defined as $\phi(t) := (2\pi)^{-1/2} e^{-t^2/2}$, $\Phi(t) := \int_{-\infty}^{t} \phi(s) ds$. The Heaviside step function $\Theta : \mathbb{R} \to \{0, 1\}$ is defined by $\Theta(x) = \mathbf{1}_{\{x \geq 0\}} = 1$ if $x \geq 0$ and 0 otherwise. For all $p \in [1, \infty]$, $\|\boldsymbol{x}\|_p := (\sum_{i=1}^{d} |x_i|^p)^{1/p}$ denotes the $\ell_p$ norm of $\boldsymbol{x} \in \mathbb{R}^d$. In particular, we denote $\ell_2$ norm as $\|\cdot\|$. $I_d \in \mathbb{R}^{d \times d}$ denotes the $d \times d$ identity matrix. $\mathbf{0}_d$ denotes the zero vector $(0, \ldots, 0)^\top \in \mathbb{R}^d$. We use asymptotic order notation with respect to a variable $d$. Specifically, for nonnegative functions $f, g$ of $d$, we define $f = \mathcal{O}_d(g)$ if there exist constants $C > 0$ and $d_0$ such that $f(d) \leq Cg(d)$ for all $d \geq d_0$. Similarly, we define $f = \Theta_d(g)$ if there exist constants $c, C > 0$ and $d_0$ such that $cg(d) \leq f(d) \leq Cg(d)$ for all $d \geq d_0$.

**Data Model.** Let $n \in \mathbb{N}$ be the sample size and $d \in \mathbb{N}$ the input dimension. We observe $n$ i.i.d. pairs, $\mathcal{D} = \{(\boldsymbol{x}_\mu, y_\mu)\}_{\mu=1}^{n} \subset \mathbb{R}^d \times \mathbb{R}$, generated according to the following linear Gaussian model:

$$\boldsymbol{x}_\mu \sim \mathcal{N}(\boldsymbol{x}_\mu; \mathbf{0}, I_d), \; y_\mu = y(\boldsymbol{x}_\mu; \boldsymbol{w}^*) = \frac{1}{\sqrt{d}} \boldsymbol{x}_\mu^\top \boldsymbol{w}^* + \xi_\mu,$$

with the ground-truth parameter $\boldsymbol{w}^* \in \mathbb{R}^d$ and independent noise $\xi_\mu \sim \mathcal{N}(\xi_\mu; 0, \sigma^2)$. The components of the parameter are independently drawn from the distribution $p_{\boldsymbol{w}^*}$ on $\mathbb{R}$, with bounded moments and satisfying $\|\boldsymbol{w}^*\|_2^2 = \rho d$. Despite its simplicity, this setting is expressive enough to capture a broad class of problems. In particular, it has been shown that asymptotic behavior of Gaussian universality regression problems, across a large class of features, can be computed to leading order under a simpler model with Gaussian features (Loureiro et al., 2021; Goldt et al., 2022; Montanari & Saeed, 2022; Gerace et al., 2024; Dandi et al., 2023).

**Quantized Linear Predictor.** Let $\boldsymbol{w} \in \mathbb{R}^d$ be a trainable parameter. We define the prediction through the following quantized linear model:

$$\hat{y}(\boldsymbol{x}; \boldsymbol{w}) = \frac{1}{\sqrt{d}} \boldsymbol{\psi}(\boldsymbol{w})^\top \boldsymbol{\psi}(\boldsymbol{x}),$$

where the quantization map $\boldsymbol{\psi} : \mathbb{R}^d \to \Omega^d$ acts component-wise as $\boldsymbol{\psi}(\boldsymbol{w}) = (\psi(w_1), \ldots, \psi(w_d))^\top$ and $\Omega = \{v_k \mid v_k \in \mathbb{R}\}_{k=0}^{L}$ denotes the finite set of quantization levels. Specifically, $\boldsymbol{\psi} : \mathbb{R}^d \to \Omega^d$ maps a real vector $\boldsymbol{w} \in \mathbb{R}^d$ to its nearest discrete counterpart $\hat{\boldsymbol{w}} \in \Omega^d$. In the uniform $b$-bit quantization scheme with range $\omega > 0$, the number of quantization levels is $L + 1 = 2^b - 1$. We define the quantization levels as $\Omega = \{v_k \mid v_k = -\omega + k\Delta\}_{k=0}^{L}$, where

the quantization step size $\Delta = 2\omega/L$. Thus, $\Omega$ uniformly partitions the interval $[-\omega, \omega]$ into $L$ equal subintervals. The corresponding decision thresholds $\{\theta\}_{k=1}^{L}$ are given by $\theta_k = -\omega + (k - 1/2)\Delta$, $k = 1, \ldots, L$, which uniquely determine the mapping of each real value $x \in \mathbb{R}$ to its nearest quantization level $v_k$. Formally, the scalar quantizer $\psi : \mathbb{R} \to \Omega$ is defined by

$$\psi(x) = -\omega + \Delta \sum_{k=1}^{L} \Theta(x - \theta_k). \tag{1}$$

We further generalize a differentiable relaxation of the quantizer, parameterized by a temperature $T \geq 0$:

$$\psi_T(x) = -\omega + \Delta \sum_{k=1}^{L} \Phi\left(\frac{x - \theta_k}{T}\right).$$

As $T \to +0$, the relaxed quantizer $\psi_T(x)$ converges pointwise to the hard quantizer defined in Eq. (1).

**Empirical Risk.** We define the empirical risk for $\boldsymbol{w} \in \mathbb{R}^d$ with regularization coefficient $\lambda > 0$ as

$$\mathcal{L}(\boldsymbol{w}; \mathcal{D}) = \sum_{\mu=1}^{n} l(\boldsymbol{w}; (\boldsymbol{x}^\mu, y^\mu)),$$

$$l(\boldsymbol{w}; (\boldsymbol{x}^\mu, y^\mu)) = \tfrac{1}{2} (y^\mu - \hat{y}(\boldsymbol{x}^\mu; \boldsymbol{w}))^2 + \tfrac{\lambda}{2d} \|\boldsymbol{\psi}(\boldsymbol{w})\|^2. \tag{2}$$

Since $\boldsymbol{\psi}$ is piecewise constant, the resulting objective function is both non-convex and non-differentiable. Although this formulation is simple, it is closely related to layer-wise PTQ methods (Frantar et al., 2022; Lin et al., 2024; Arai & Ichikawa, 2025; Zhao et al., 2025), which are among the most widely adopted approaches for quantizing large-scale language models. These methods achieve state-of-the-art performance by solving layer-wise optimization problems that are structurally equivalent to Eq. (2). A more detailed discussion of this relationship is provided in Supplement F. Therefore, characterizing the quantization properties in this setting has the potential to enable the development of more effective layer-wise PTQ algorithms.

**Straight Through Estimator.** Training with quantization is challenging because discrete low-precision weights and inputs make the loss function piecewise constant, thus prohibiting the direct use of standard backpropagation. A widely used approach is the straight-through estimator (STE) (Bengio et al., 2013), which enables gradient-based optimization by employing surrogate gradients for piecewise constant objectives. STE is an empirically effective heuristic that, during the backward pass only, replaces the almost-everywhere-zero derivative of discrete components

with surrogate Jacobian. Specifically, the following approximation is employed using the chain-rule approximation:

$$\boldsymbol{w}^{t+1} = \boldsymbol{w}^t - \eta\left(\frac{\partial l(\boldsymbol{w}^t; (\boldsymbol{x}^t, y^t))}{\partial \boldsymbol{\psi}(\boldsymbol{w}^t)}\right)\left(\frac{\partial \boldsymbol{\psi}(\boldsymbol{w}^t)}{\partial \boldsymbol{w}^t}\right)$$
$$\approx \boldsymbol{w}^t - \eta\left(\frac{\partial l(\boldsymbol{w}^t; (\boldsymbol{x}^t, y^t))}{\partial \boldsymbol{\psi}(\boldsymbol{w}^t)}\right).$$

To simplify the theoretical analysis, we consider a one-pass STE dynamics in which each data sample is used only once. Specifically, at step $t$, the parameter $\boldsymbol{w}^t$ is updated with a new sample $(\boldsymbol{x}^t, y^t)$ as follows:

$$\boldsymbol{w}^{t+1} = \boldsymbol{w}^t - \eta\left[\frac{(\hat{y}(\boldsymbol{x}^t; \boldsymbol{w}) - y^t)}{\sqrt{d}}\boldsymbol{\psi}(\boldsymbol{x}^t) + \frac{\lambda}{d}\boldsymbol{\psi}(\boldsymbol{w}^t)\right],$$

where $\eta \in \mathbb{R}_+$ denotes the learning rate. The SGD algorithm induces a Markov process $(\boldsymbol{w}_t)_{t \in [T]}$, where $T$ is the number of steps. Note that our analysis can naturally extend to mini-batch STE where the mini-batch size remains finite, i.e., $\mathcal{O}_d(1)$.

**Generalization Metric.** We focus on the generalization error, which quantifies the performance of the current iterate $\boldsymbol{w}^t$. Given a newly sampled pair $(\boldsymbol{x}_{\text{new}}, y_{\text{new}})$ from the same distribution as the dataset $\mathcal{D}$, we define the generalization error as

$$\varepsilon_g^t = \mathbb{E}_{(y_{\text{new}}, \boldsymbol{x}_{\text{new}})}\left[\left(y_{\text{new}} - \hat{y}(\boldsymbol{x}_{\text{new}}; \boldsymbol{w}^t)\right)^2\right],$$

where $\mathbb{E}_{(\boldsymbol{x}_{\text{new}}, y_{\text{new}})}[\cdot]$ denotes expectation with respect to the data distribution. A straightforward calculation gives

$$\varepsilon_g^t = \sigma_\psi^2 q_\psi^t - 2\kappa_\psi m_\psi^t + \rho + \sigma^2, \tag{3}$$

where

$$q_\psi^t = \frac{\|\boldsymbol{\psi}(\boldsymbol{w}^t)\|_2^2}{d}, \quad m_\psi^t = \frac{\boldsymbol{\psi}(\boldsymbol{w}^t)^\top \boldsymbol{w}^*}{d}.$$

We further define $r_\psi^t = \boldsymbol{\psi}(\boldsymbol{w}^t)^\top \boldsymbol{w}^t/d$. In this paper, we refer to such statistics derived from $\boldsymbol{w}^t$ as macroscopic states, which are commonly called "order parameters" in statistical mechanics. The constants $\sigma_\psi^2$ and $\kappa_\psi$ depend only on the quantizer $\psi$ and the input distribution:

$$\sigma_\psi^2 = \mathbb{E}_{x \sim \mathcal{N}(0,1)}[\psi(x)^2] = \sum_{i=0}^L v_i^2(\Phi(\theta_{i+1}) - \Phi(\theta_i)),$$

$$\kappa_\psi = \mathbb{E}_{x \sim \mathcal{N}(0,1)}[x\psi(x)] = \sum_{i=1}^L (v_i - v_{i-1})\phi(\theta_i).$$

The detailed derivation is provided in Supplement B.

## 4. Microscopic Dynamics

This section derives a microscopic description in which the empirical distribution of the coordinates of the learnable parameter $\boldsymbol{w}^t$ satisfies a partial differential equation (PDE). Rather than tracking the full trajectory of the high-dimensional vector $\boldsymbol{w}^t$, we focus on the following empirical distribution.

**Definition 4.1.** For $t \in \mathbb{N}$, let $w_i^t$ denote the $i$th coordinate of $\boldsymbol{w}^t$, and let $w_i^*$ the corresponding coordinate of the ground-truth vector $\boldsymbol{w}^*$. We define the empirical measure, given by

$$\mu_t(\hat{w}^*, \hat{w}) := \frac{1}{d}\sum_{i=1}^d \delta(\hat{w}^* - w_i^*)\delta(\hat{w} - w_i^t).$$

We embed the discrete-time process into a continuous-time one by defining $\mu_\tau^{(d)} := \mu_{\lfloor \tau \times d \rfloor}$ with $t = \lfloor \tau \times d \rfloor$. Following the general approach of (Wang et al., 2017; 2019), we establish our results under the following assumptions.

*Assumption* 4.2. We define the macroscopic state $\Psi^t = (m^t, q^t) \in \mathbb{R}^2$ by $m^t = \boldsymbol{w}^{*\top}\boldsymbol{w}^t/d$, $q^t = \|\boldsymbol{w}^t\|^2/d$.

(1) The pairs $(\boldsymbol{x}^t, \xi^t)_{t \in [T]}$ are i.i.d. random variables across $t \in [T]$.

(2) The initial macroscopic state $\Psi^0$ satisfies $\mathbb{E}\|\Psi^0 - \bar{\Psi}^0\| \leq C/\sqrt{d}$, where $\bar{\Psi}^0 \in \mathbb{R}^2$ is a deterministic vector and $C$ is a constant independent of $d$.

(3) The fourth moments of the initial parameter $\boldsymbol{w}^0$ are uniformly bounded: $\mathbb{E}\sum_{i=1}^d (w_i^*)^4 + (w_i^0)^4 \leq C$ where $C$ is a constant independent of $d$.

Assumption (A.1) for $(\boldsymbol{x}^t, \xi^t)$ can be relaxed to non-Gaussian cases if all moments are bounded; however, we use the Gaussian assumption to simplify the proof. Assumption (A.2) ensures that the initial macroscopic states concentrate around deterministic values. Assumption (A.3) requires the coordinates of the ground-truth vector and the initial parameters to be $\mathcal{O}_d(1)$. Then, we can show that the following theorem holds:

**Theorem 4.3.** *Under Assumption 4.2, for any finite $T > 0$, the empirical measure $\mu_t^{(d)}$ converges weakly to a process $\mu_\tau$, which is the law of the solution to the stochastic differential equation:*

$$\mathrm{d}\hat{w}_\tau = \eta\big(\kappa_\psi \hat{w}^* - (\sigma_\psi^2 + \lambda)\psi_T(\hat{w}_\tau)\big)\mathrm{d}\tau + \eta\sigma_\psi\varepsilon_g^{\frac{1}{2}}(\tau)\mathrm{d}B_\tau,$$

*where $(\hat{w}^*, \hat{w}_0) \sim \mu_0$; $B_t$ is the standard Brownian motion; $\varepsilon_g(\tau)$ defined by*

$$\varepsilon_g(\tau) = \sigma_\psi^2 q_\psi(\tau) - 2\kappa_\psi m_\psi(\tau) + \rho + \sigma^2,$$

*where $q_\psi(\tau) = \mathbb{E}_{\mu_\tau}[\psi(\hat{w}_\tau)^2]$, $m_\psi(\tau) = \mathbb{E}_{\mu_\tau}[\psi(\hat{w}_\tau)\hat{w}^*]$.*

The detailed derivation is provided in Supplement C. By Itô integral formula, the deterministic measure $\mu_t$ is the unique solution to the following PDE: for any bounded smooth test function $\zeta(\hat{w}^*, \hat{w}_\tau)$,

$$\frac{\mathrm{d}}{\mathrm{d}\tau}\mathbb{E}_{\mu_\tau}[\zeta] = \mathbb{E}_{\mu_\tau}\left[\eta(\kappa_\psi\hat{w}^* - (\sigma_\psi^2 + \lambda)\psi_T(\hat{w}_\tau))\,\partial_{\hat{w}}\zeta\right]$$
$$+ \frac{\eta^2}{2}\sigma_\psi^2\varepsilon_g(\tau)\mathbb{E}_{\mu_\tau}[\partial_{\hat{w}}^2\zeta], \quad (4)$$

Therefore, at the level of measure,

$$\frac{\partial\mu_\tau}{\partial\tau} = -\frac{\partial}{\partial\hat{w}_\tau}\left(\eta(\kappa_\psi\hat{w}^* - (\sigma_\psi^2 + \lambda)\psi_T(\hat{w}))\mu_\tau\right)$$
$$+ \frac{\eta^2}{2}\sigma_\psi^2\varepsilon_g(\tau)\frac{\partial^2\mu_\tau}{\partial\hat{w}^2}. \quad (5)$$

We refer readers to (Wang et al., 2017; 2019) for a general framework establishing the above scaling limit.

**Numerical Validation.** We validate the predictions of Eq. (5) by considering the vector $\boldsymbol{w}^* = \mathbf{1}_d$ in dimension $d = 3{,}000$. The initial distribution is given by $\mu_0(\hat{w}|\hat{w}^* = 1) = \mathcal{N}(0,1)$. In all simulations, both the inputs and weights are quantized in the zero-temperature limit ($T \to +0$) with $\omega = 2$ and bit-width $b = 2$. Figure 1 shows the probability distribution $\mu_\tau(\hat{w}|\hat{w}^* = 1)$. The prediction of Eq. (5) agrees closely with the simulation results. Furthermore, as $\tau$ increases, the distribution approaches $\hat{w}^* = 1$.

## 5. Macroscopic Dynamics

As shown in Eq. (3), the generalization error $\varepsilon_g^t$ is a function of $q_\psi^t$, $m_\psi^t$. Our goal is therefore to characterize $\varepsilon_g^t$ by analyzing the time evolution of the macroscopic states. To connect the microscopic and macroscopic descriptions, we begin with the PDE in Eq. (4). By selecting test functions $\zeta(\hat{w}^*, \hat{w})$ corresponding to $\psi(\hat{w})^2$ and $\hat{w}^*\psi(\hat{w})$, and substituting them into Eq. (4), we evaluate evolution equations for $q_\psi(\tau)$ and $m_\psi(\tau)$. However, the resulting equations involve additional nonlinear expectations, such as $\zeta(\hat{w}^*, \hat{w}) = \psi_T(\hat{w})\psi_T'(\hat{w}), \hat{w}^*\psi_T'(\hat{w}), \psi_T'\psi''(\hat{w})$, among others. The detailed explanation is provided in Supplement C. Although the PDE can in principle be solved numerically to evaluate the generalization error in the high-dimensional limit, this strategy offers limited analytical insight for subsequent investigations such as fixed-point and stability analysis.

**Isotropy in Orthogonal Directions.** To obtain a closed and tractable system, we consider relating the macroscopic states $\Psi^t$ to the nonlinear macroscopic ones $m_\psi^t, q_\psi^t$. To this end, we decompose $\boldsymbol{w}^t = \boldsymbol{w}_\parallel^t + \boldsymbol{w}_\perp^t$, where $\boldsymbol{w}_\parallel^t$ is parallel to the ground truth vector $\boldsymbol{w}^*$ and $\boldsymbol{w}_\perp^t$ lies in the orthogonal subspace. Algebraically, we have $\|\boldsymbol{w}_\perp^t\|^2 = d(q_t - m_t^2\rho^{-1})$.

In the high-dimensional limit, SGD cannot distinguish directions within the orthogonal subspace under isotropic inputs; except for the ground-truth direction, the dynamics are approximately rotationally symmetric. We therefore assume the orthogonal component to be isotropic Gaussian in the $d \to \infty$ limit.

*Assumption 5.1.* For any $t \in [T]$,

$$p(\boldsymbol{w}_\perp^t) \underset{d\to\infty}{\Longrightarrow} \mathcal{N}\left(\mathbf{0}_d, (q_t - m_t^2\rho^{-1})I_d\right).$$

Under Assumption 5.1, the nonlinear macroscopic states can be expressed explicitly as functions of $\Psi^t$.

*Proposition 5.2.* Let $s^t = (q^t - (m^t)^2/\rho)^{1/2}$. Under Assumption 5.1, for any $t \in [T]$,

$$m_\psi(m^t, s^t) = \mathbb{E}_{w^*}\left[w^*\left(-\omega + \Delta\sum_{i=1}^{L}\Phi\left(\frac{\frac{m^t w^*}{\rho} - \theta_i}{s^t}\right)\right)\right],$$

$$q_\psi(m^t, s^t) = \mathbb{E}_{w^*}\left[v_0^2 + \sum_{i=1}^{L}(v_i^2 - v_{i-1}^2)\Phi\left(\frac{\frac{m^t w^*}{\rho} - \theta_i}{s^t}\right)\right],$$

$$r_\psi(m^t, s^t) = \frac{m^t m_\psi}{\rho} + \Delta s^t\sum_{i=1}^{L}\mathbb{E}_{w^*}\left[\phi\left(\frac{\frac{m^t w^*}{\rho} - \theta_i}{s^t}\right)\right],$$

Once the dynamics of $\Psi^t$ are evaluated, the dynamics of $m_\psi$, $q_\psi$ can be analyzed. The details are provided in Supplement D.1.

**Concentration to ODEs.** Combining Assumption 4.2 and 5.1 yields the following result.

**Theorem 5.3.** *For all $T > 0$, under Assumptions 4.2 and 5.1, we have*

$$\max_{0\leq t\leq d\times T}\mathbb{E}\|\Psi^t - \Psi(t/d)\|_2 \leq Cd^{-\frac{1}{2}},$$

*where $C$ is a constant independent of $d$ and $\Psi(\tau) = (m(\tau), q(\tau))$ is a unique solution to*

$$\frac{\mathrm{d}m(\tau)}{\mathrm{d}\tau} = -\eta\left((\sigma_\psi^2 + \lambda)m_\psi(\tau) - \kappa_\psi\rho\right), \quad (6)$$

$$\frac{\mathrm{d}q(\tau)}{\mathrm{d}\tau} = -2\eta\left((\sigma_\psi^2 + \lambda)r_\psi(\tau) - \kappa_\psi m(\tau)\right) + \eta^2\sigma_\psi^2\varepsilon_g(\tau),$$
$$(7)$$

*where $\varepsilon_g(\tau) = \sigma_\psi^2 q_\psi(\tau) - 2\kappa_\psi m_\psi(\tau) + \rho + \sigma^2$, and initial condition $\Psi(0) = \bar{\Psi}$.*

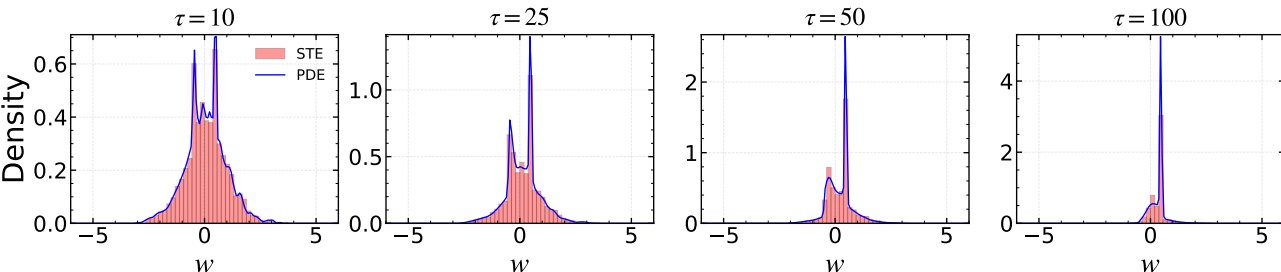

*Figure 1.* Comparison between STE simulations (red histograms) and the PDE prediction (blue curves) for the probability density $\mu_\tau(\hat{w}|w^* = 1)$ at training horizons $\tau \in \{10, 25, 50, 100\}$. Simulations use $d = 3{,}000$, $\boldsymbol{w}^* = \mathbf{1}$, and $\mu_0$ the standard Gaussian. Both the inputs and weights are quantized with $\omega = 2$ and bit-width $b = 2$.

The detailed derivation is given in Supplement D. The result can be proved using standard convergence techniques for stochastic processes, together with a coupling trick (Wang et al., 2018; 2019; Ichikawa & Hukushima, 2024). In brief, the one-step increment can be decomposed into a conditional drift term and a martingale increment as follows:

$$\Psi^{t+1} - \Psi^t = \mathbb{E}_t \Psi^{t+1} - \Psi^t + \left( \Psi^{t+1} - \mathbb{E}_t \Psi^{t+1} \right),$$

where $\mathbb{E}_t$ denotes conditional expectation given the current state of the Markov chain $\boldsymbol{w}^t$. Thus, it suffices to verify, for all $t \le dT$, that

$$\mathbb{E}\|\mathbb{E}_t \Psi^{t+1} - \Psi^t - F(\Psi^t)/d\| \le C d^{-2/3},$$
$$\mathbb{E}\|\Psi^{t+1} - \mathbb{E}_t \Psi^{t+1}\|^2 \le C d^{-2},$$

where $F$ denotes the right-hand sides of Eqs. (25)- (26). The first bound ensures that the leading-order drift is captured by the ODEs in Theorem 5.3, while the second bound guarantees that stochastic fluctuations vanish as $d \to \infty$.

## 6. Results

This section evaluates the learning dynamics predicted by the ODEs. We also verify the theoretical prediction using STE simulations with finite dimensions. Additional ablation studies are provided in Supplement G.

### 6.1. Weight-Only Quantization

**Dependence on Bit Width.** We examine how the bit width $b$ influences the trajectory of the generalization error. In the representative setting of $\eta = 0.04$, $\lambda = 1$, $\omega = 1$, and $T = 0$, $\boldsymbol{w}^* = \mathbf{1}_d$, we compare the ODE under the isotropy assumption with STE simulations at finite dimension of $d = 900$. Figure 2 shows a close agreement across all bit widths. A characteristic two-stage behavior emerges: after an initial plateau, $\varepsilon_g$ drops abruptly on a bit-width-dependent timescale and then settles into a stationary regime, with its floor decreasing as $b$ increases. A quantitative fixed-point analysis, comparing the quantized and unquantized

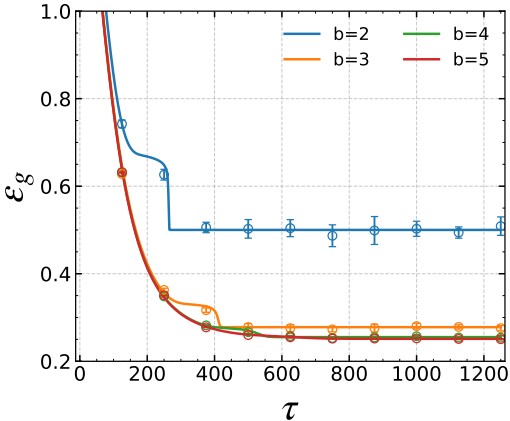

*Figure 2.* Generalization error $\varepsilon_g$ as a function of training time $\tau$ for bit widths $b \in \{2, 3, 4, 5\}$, with $\eta = 0.04$, $\lambda = 1$, $\omega = 1.0$, and $d = 900$. Symbols with error bars indicate STE simulations averaged over five independent runs; solid curves indicate the ODE prediction.

models as a function of the quantization hyperparameters $b$ and $\omega$, is presented in Section 6.3. We next fix $b$ and demonstrate how the quantization range $\omega$ influences the onset and depth of the transition.

**Dependence on Quantization Range.** Fixing $b = 3$ as a representative case, we vary the quantization range $\omega$ while keeping $\eta = 0.04$, $\lambda = 1$, $T = 0$, and $d = 900$ fixed. Figure 3 shows that both the convergence rate and the occurrence and timing of the abrupt drop depend sensitively on $\omega$: small ranges, e.g., $\omega = 0.25$, slow convergence and raise the error floor, whereas moderate to large ranges accelerate the transition and yield lower generalization error. These results suggest that careful calibration of $\omega$ matters not only for quantization fidelity but also for favorable STE dynamics, further motivating learning the range/scale parameter rather than fixing it, as proposed by Cheng et al. (2024).

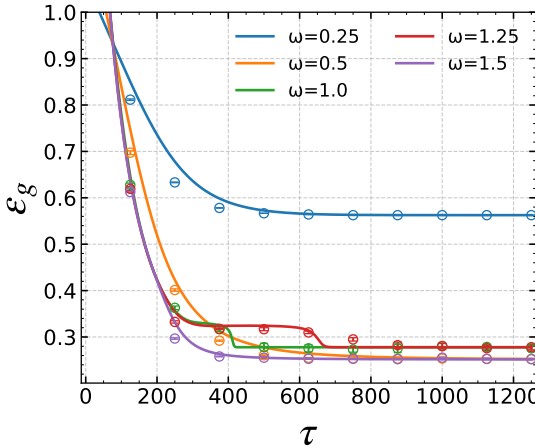

*Figure 3.* Generalization error $\varepsilon_g$ as a function of training time $\tau$ at fixed bit-width $b = 3$, for quantization ranges $\omega \in \{0.25, 0.50, 1.00, 1.25, 1.50\}$. Symbols with error bars denote STE simulations averaged over five runs; solid curves denote the ODE prediction.

## 6.2. Weight-Input Quantization

We next examine the effect of input quantization. As a representative case, we use $\omega = 1.0$, $b \in \{3, 4\}$, $T = 0$, $\eta = 0.05$, and $\lambda = 1$. We vary the input bit width $b_x \in \{3, 4, 5\}$ while fixing the input range to $\omega_x = 1$. Figure 4 shows that the ODE tracks the STE simulation with dimension $d = 500$ closely. Input quantization markedly degrades performance relative to unquantized inputs: the error floor rises and convergence slows. As expected, increasing $b_x$ improves performance; however, in this minimal setting the gains beyond moderate precision are incremental. At $b_x = 3$, we also observe non-monotone transients, delayed or oscillatory trajectories before settling. While commodity hardware often uses the same bit widths (e.g., 4-bit weights $\times$ 4-bit inputs), these results suggest that higher input precision yields faster convergence and a lower generalization error, whereas low-bit input may slow learning due to the non-monotone dynamics as shown in Figure 4.

## 6.3. Fixed–Point Analysis

The learning curves reveal finite-time behavior and not characterize the long-time behavior $\tau \to \infty$: where the dynamics settle, which values the limiting states take, and how stability depends on quantization. To study this regime, we analyze the fixed points. A fixed point is *locally asymptotically stable* if the Jacobian at the point has its spectrum strictly in the left half–plane, and *marginally stable* if the spectrum lies in the closed left half–plane and includes zero.

**Input–Only Quantization.** We first isolate the effect of input quantization while keeping the weights real–valued. In this setting, both the fixed point and its stability admit

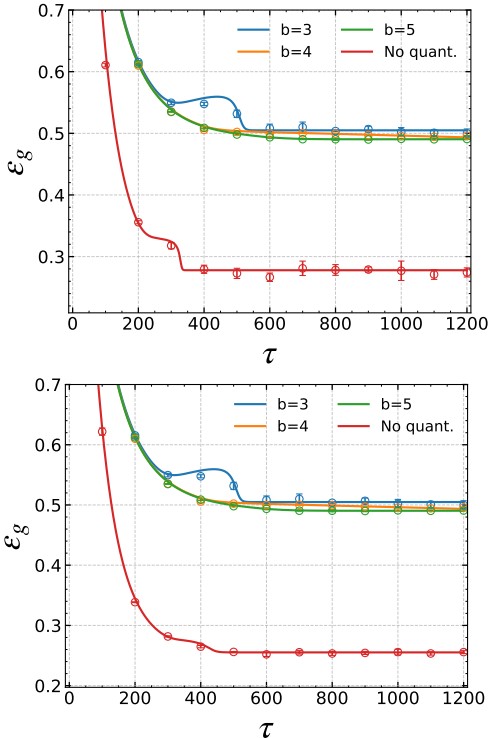

*Figure 4.* Joint quantization of *weights* and *inputs* with the range $\omega = 1.0$ and $b \in \{3, 4\}$. Training time $\tau$-dependence of generalization error $\varepsilon_g$ at $T = 0$, $d = 500$, $\eta = 0.05$, $\lambda = 1.0$ for input bit widths $b_x \in \{3, 4, 5\}$, with an unquantized-input one ("No quant."). Symbols with error bars denote STE simulations averaged over five runs; solid curves show the ODE prediction. Top panel: $b = 3$; bottom panel: $b = 4$.

closed–form expressions.

*Proposition* 6.1. If the learning rate satisfies

$$0 < \eta < 2(\sigma_\psi^2 + \lambda)/\sigma_\psi^4,$$

then the macroscopic ODE has a locally asymptotically stable fixed point $(m^*, q^*)$ with generalization error

$$\varepsilon_g^* = \rho + \sigma^2 + \sigma_\psi^2 q^* - 2\kappa_\psi m^*,$$

where $m^* = \rho \kappa_\psi / (\sigma_\psi^2 + \lambda)$ and

$$q^* = \frac{2\kappa_\psi^2 + \eta\, \sigma_\psi^2 \big((\rho + \sigma^2)(\sigma_\psi^2 + \lambda) - 2\kappa_\psi^2\big)}{(\sigma_\psi^2 + \lambda)\big(2(\sigma_\psi^2 + \lambda) - \eta\sigma_\psi^4\big)}.$$

Details are provided in Supplement E.1. Setting $\kappa_\psi = \sigma_\psi^2 = 1$ reproduces the unquantized linear regression baseline. Figure 5 focuses on the noiseless, unregularized case, i.e., $\lambda = 0$, $\sigma^2 = 0$ and presents two complementary views as functions of the input quantization range $\omega_x$ for bit widths $b_x \in \{2, 3, 4, 10\}$. The left panel shows the stability boundary $2/\sigma_\psi^2$, above which no asymptotically stable fixed point exists. Contrary to the naive expectation that lower precision is always less stable, the stability margin is not monotonic

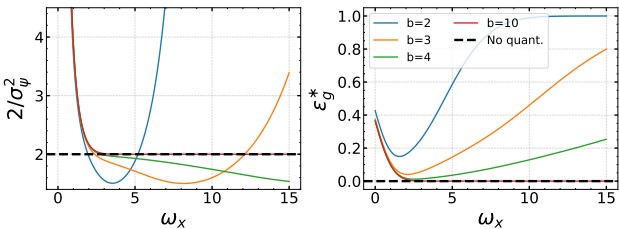

*Figure 5.* Long-time behavior under input-only quantization with $\lambda = 0$ and $\sigma^2 = 0$. Left: Stability boundary at $2/\sigma_\psi^2$; for $\eta > 2/\sigma_\psi^2$, no asymptotically stable fixed point exists. The dashed line indicates the unquantized baseline. Right: Steady-state generalization error $\varepsilon_g^*$.

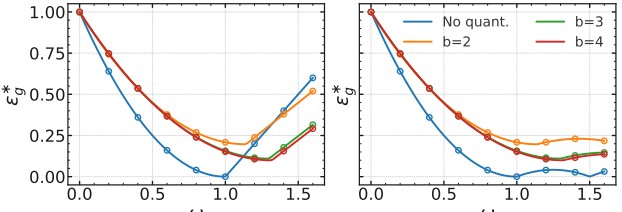

*Figure 6.* Dependence of the generalization error $\varepsilon_g^*$ on $\omega$ for input quantizers with $b_x \in \{2, 3, 4\}$ and $\omega_x = 1$, including the unquantized baseline. Left: weight quantization $b = 2$; Right: $b = 3$. All curves use identical settings across panels: regularization $\lambda = 0$, noise variance $\sigma^2 = 0$, STE simulation with $d = 100$ and learning rate $\eta = 10^{-4}$, run up to $8 \times 10^6$.

in bit width. In some ranges, the stable region is larger than in unquantized linear regression, implying that the quantization range acts as an implicit regularizer.

**Joint Input–Weight Quantization.** When both inputs and weights are quantized, the fixed-point equations are nonlinear:

$$m_\psi(m^*, s^*) = \frac{\kappa_\psi \rho}{\sigma_\psi^2 + \lambda}, \qquad (8)$$

$$\frac{2(\sigma_\psi^2 + \lambda)}{\sigma_\psi^2} s^* \sum_{k=1}^{L-1} \phi\left(\frac{m^* - \theta_k}{s^*}\right) = \eta \varepsilon_g(m^*, s^*).$$

Closed-form solutions are not available, hence we focus on the small-learning-rate limit. From Eq. (8), the solution $m^*$ is unique; details are provided in E.2. Based on this result, we introduce the following definition.

**Definition 6.2.** Let $c = \kappa_\psi \rho / (\sigma_\psi^2 + \lambda)$. Define the unique index $i^* \in \{0, \ldots, L-1\}$ by $v_{i^*} \leq c \leq v_{i^*+1}$, and define the fractional position $p := (c - v_{i^*})/\Delta \in [0, 1]$.

**Theorem 6.3** (Informal)**.** *In the small-learning-rate limit,*

$$\varepsilon_g^* = \begin{cases} \varepsilon_g^{(0)} + \sigma_\psi^2 \Delta^2 p(1-p) + o(\eta), & |c| < \omega, p \in (0, 1), \\ \varepsilon_g^{(0)} + o(1/\sqrt{\log(1/\eta)}), & |c| < \omega, p \in \{0, 1\}, \\ \rho + \sigma^2 - 2\kappa_\psi \omega + \sigma_\psi^2 \omega^2 + o(\eta), & |c| \geq \omega, \end{cases}$$

*where $\varepsilon_g^{(0)} = \rho + \sigma^2 - 2\kappa_\psi c + \sigma_\psi^2 \omega^2$ denotes the generalization error of the input-only quantized model in the small-learning-rate limit.*

The detailed derivation is provided in Supplement E.2. Figure 6 shows the dependence of the generalization error $\varepsilon_g^*$ on $\omega$, with the input quantizer fixed at $\omega_x = 1$ and $b_x \in \{2, 3, 4\}$ including the unquantized case, while the weight quantizer uses $b \in \{2, 3\}$. The results include STE simulations with $d = 100$, a learning rate of $\eta = 10^{-4}$, and a training duration of up to $\tau = 8 \times 10^6$. These results indicate that the fixed point is also non-monotonic in $\omega$, and that the optimal value is contingent on both the weight and input quantizers. The leading correction is $\mathcal{O}_\Delta(\Delta^2)$

and varies non-monotonically with $p$, thus quantifying how weight quantization perturbs the input-quantized fixed point.

## 7. Conclusion

We develop a high-dimensional framework for jointly quantizing *weights* and *inputs*, trained using STE. Our analysis reveals that STE dynamics are dependent on key hyperparameters: *bit width* and *quantization range*. We identify several phenomena: (i) a typical two-phase trajectory consisting of an initial plateau followed by a sharp drop in generalization error, and (ii) convergence delays caused by non-monotonic transients induced by low-bit input quantization. Quantization can expand the stable learning region, allowing higher learning rates and acting as an implicit regularizer rather than merely as noise. Methodologically, we extend conventional high-dimensional learning-dynamics analyses to settings with nonlinear transformations of both parameters and inputs. This method is not limited to quantizers; it also applies to a broader class of nonlinear transformations, such as weight normalization. We aim to scale the framework to multilayer architectures and structured data to show how architectural redundancy and depth interact with STE dynamics.

## Impact Statement

This paper presents work whose goal is to advance the field of Machine Learning. There are many potential societal consequences of our work, none which we feel must be specifically highlighted here.

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

# A. Useful Lemmas

We collect Gaussian convolution identities used repeatedly below. Throughout, let $X \sim \mathcal{N}(m, s^2)$, $s > 0$, let $Z \sim \mathcal{N}(0,1)$, and let $a \in \mathbb{R}$, and $T > 0$. All standard normal variables introduced below are assumed independent of $X$.

*Lemma* A.1.

$$\mathbb{E}\left[\Phi\left(\frac{X-a}{T}\right)\right] = \Phi\left(\frac{m-a}{\sqrt{s^2+T^2}}\right).$$

*Proof.* Since $\Phi(u) = \mathbb{P}[Z \leq u]$,

$$\mathbb{E}\left[\Phi\left(\frac{X-a}{T}\right)\right] = \mathbb{P}\left(Z \leq \frac{X-a}{T}\right) = \mathbb{P}[X - TZ \geq a].$$

The random variable $X - TZ$ is Gaussian with mean $m$ and variance $s^2 + T^2$, so the right-hand side equals $\Phi((m-a)/\sqrt{s^2+T^2})$.
$\square$

*Lemma* A.2.

$$\mathbb{E}\left[\phi\left(\frac{X-a}{T}\right)\right] = \frac{T}{\sqrt{s^2+T^2}}\phi\left(\frac{m-a}{\sqrt{s^2+T^2}}\right).$$

*Proof.* By definition,

$$\begin{aligned}
\mathbb{E}\left[\phi\left(\frac{X-a}{T}\right)\right] &= \int_{-\infty}^{\infty} \frac{1}{\sqrt{2\pi}s} e^{-\frac{(x-m)^2}{2s^2}} \frac{1}{\sqrt{2\pi}} e^{-\frac{(x-a)^2}{2T^2}} \\
&= \frac{1}{2\pi s} e^{-\frac{(m-a)^2}{2(s^2+T^2)}} \int_{-\infty}^{\infty} e^{-\frac{(x-\mu)^2}{2(s^2+T^2)}} \, dx \\
&= \frac{1}{2\pi s} e^{-\frac{(m-a)^2}{2(s^2+T^2)}} \sqrt{2\pi}\sqrt{s^2+T^2} \\
&= \frac{T}{\sqrt{s^2+T^2}}\phi\left(\frac{m-a}{\sqrt{s^2+T^2}}\right).
\end{aligned}$$

$\square$

*Lemma* A.3.

$$\mathbb{E}\left[\Phi\left(\frac{X-a}{T}\right)\Phi\left(\frac{X-b}{T}\right)\right] = \Phi_2\left(\frac{m-a}{\sqrt{s^2+T^2}}, \frac{m-b}{\sqrt{s^2+T^2}}; \frac{s^2}{s^2+T^2}\right),$$

where $\Phi_2(\cdot, \cdot; \rho)$ denotes the standard bivariate normal CDF with correlation $\rho$.

*Proof.* Let $Z_1, Z_2 \sim \mathcal{N}(0,1)$ be independent and independent of $X$. Then

$$\mathbb{E}\left[\Phi\left(\frac{X-a}{T}\right)\Phi\left(\frac{X-b}{T}\right)\right] = \mathbb{P}[X - TZ_1 \geq a, X - TZ_2 \geq b],$$

We have

$$\mathbb{E}[X - TZ_1 - a] = m - a, \;\; \text{Var}[X - TZ_1 - a] = s^2 + T^2, \;\; \text{Cov}[X - TZ_1 - a, X - TZ_2 - b] = s^2.$$

Thus, the following variable

$$\left(\frac{X - TZ_1 - a}{\sqrt{s^2+T^2}}, \frac{X - TZ_2 - b}{\sqrt{s^2+T^2}}\right),$$

is bivariate standard normal distribution with means $(m-a/\sqrt{s^2+T^2}, m-b/\sqrt{s^2+T^2})$ and correlation $\rho = s^2/(s^2+T^2)$. Therefore,

$$\mathbb{P}\left[\frac{X - TZ_1 - a}{\sqrt{s^2+T^2}} \geq 0, \frac{X - TZ_2 - b}{\sqrt{s^2+T^2}} \geq 0\right] = \Phi_2\left(\frac{m-a}{\sqrt{s^2+T^2}}, \frac{m-b}{\sqrt{s^2+T^2}}; \frac{s^2}{s^2+T^2}\right).$$

$\square$

*Lemma* A.4.

$$\mathbb{E}\left[X\Phi\left(\frac{X-a}{T}\right)\right] = m\Phi\left(\frac{m-a}{\sqrt{s^2+T^2}}\right) + \frac{s^2}{\sqrt{s^2+T^2}}\phi\left(\frac{m-a}{\sqrt{s^2+T^2}}\right).$$

*Proof.* By Stein's lemma,

$$\mathbb{E}\left[(X-m)\Phi\left(\frac{x-a}{T}\right)\right] = s^2\mathbb{E}\left[\frac{1}{T}\phi\left(\frac{X-a}{T}\right)\right].$$

Hence,

$$\mathbb{E}[Xf(X)] = m\mathbb{E}\left[\Phi\left(\frac{x-a}{T}\right)\right] + \frac{s^2}{T}\mathbb{E}\left[\phi\left(\frac{X-a}{T}\right)\right].$$

Substituting Lemma A.1 and A.2 completes the proof. □

*Lemma* A.5 (Mill's Inequality). For $x > 0$,

$$1 - \Phi(x) \le \frac{\phi(x)}{x}, \quad \Phi(-x) \le \frac{\phi(x)}{x}.$$

*Proof.* Since $\phi'(u) = -u\phi(u)$, integration by parts yields

$$1 - \Phi(x) = \int_x^\infty \phi(u)du = \left[-\frac{\phi(u)}{u}\right]_x^\infty - \int_x^\infty \frac{\phi(u)}{u^2}du \le \frac{\phi(x)}{x}.$$

The second inequality follows from the symmetry $\Phi(-x) = 1 - \Phi(x)$ and $\phi(-x) = \phi(x)$. □

# B. Proof of Local Field and Generalization Error

In this section, we establish two ingredients used throughout the analysis of the quantized linear model: (i) closed forms for the basic input-quantizer moments $\kappa_\psi = \mathbb{E}[X\psi(X)]$ and $\sigma_\psi^2 = \mathbb{E}[\psi(X)^2]$, and (ii) the joint asymptotic Gaussianity of the local fields that appear in the one-pass STE dynamics when both the weights and inputs are quantized. We conclude with a concise expression for the generalization error.

## B.1. Quantizer Moments

We start by deriving closed forms for the first two moments that characterize the macroscopic ODE.

*Proposition* B.1. Let $\psi : \mathbb{R} \to \{v_0, \dots, v_L\}$ be defined by

$$\psi(x) = v_k \quad \text{for } x \in [\theta_k, \theta_{k+1}),$$

with thresholds

$$\theta_k = -\omega + \left(k - \frac{1}{2}\right)\Delta, \quad k = 1, \dots, L,$$

and end points $\theta_0 = -\infty$, $\theta_{L+1} = +\infty$. Define

$$\kappa_\psi := \mathbb{E}[X\psi(X)], \quad \sigma_\psi := \mathbb{E}[\psi(X)^2].$$

Then

$$\kappa_\phi = \sum_{k=1}^L (v_{k+1} - v_k)\phi(\theta_k), \quad \sigma_\psi^2 = \sum_{k=1}^L v_k^2 p_k, \quad p_k = \Phi(\theta_{k+1}) - \Phi(\theta_k).$$

In particular, if the quantizer is symmetric around the origin, i.e., $v_k = -v_{L-k}$ and $\theta_k = -\theta_{L+1-k}$, then $\mathbb{E}[\psi(X)] = 0$ and $\text{Var}[\psi(X)] = \sigma_\psi^2$.

*Proof.* Since $\psi$ is piecewise constant

$$\mathbb{E}[X\psi(X)] = \sum_{k=0}^L v_k \mathbb{E}\left[X\mathbf{1}_{\theta_k \le X < \theta_{k+1}}\right].$$

For $a < b$, $\mathbb{E}[X\mathbf{1}_{a \leq X < b}] = \phi(a) - \phi(b)$. Then

$$\kappa_\psi = \sum_{k=0}^{L} v_k(\phi(\theta_k) - \phi(\theta_{k+1})) = \sum_{k=1}^{L}(v_k - v_{k-1})\phi(\theta_k),$$

using $\phi(\pm\infty) = 0$. Similarly,

$$\sigma_\psi^2 = \mathbb{E}[\psi(X)^2] = \sum_{k=0}^{L} v_k^2 \mathbb{P}[\theta_k \leq X < \theta_{k+1}] = \sum_{k=0}^{L} v_k^2(\Phi(\theta_{k+1}) - \Phi(\theta_k)).$$

If the quantizer is symmetric, $\mathbb{E}[\psi(X)] = 0$ by symmetry of $X$, and $\mathrm{Var}[\psi(X)] = \sigma_\psi^2$. $\qquad\square$

## B.2. Local Fields

We next formalize the high-dimensional "local fields" that appear in the stochastic updates. They obey a joint central limit theorem whose covariance is expressed in terms of the macroscopic order parameters and the two input–quantizer moments computed.

Let $\boldsymbol{x} \sim \mathcal{N}(\mathbf{0}_d, I_d)$ with independent coordinates. Define

$$\rho_d = \frac{1}{d}\|\boldsymbol{w}^*\|^2, \quad q_d = \frac{1}{d}\|\boldsymbol{w}\|^2, \quad m_d = \frac{1}{d}(\boldsymbol{w}^*)^\top \boldsymbol{w},$$

and their quantized weight counterparts

$$m_{\psi,d} = \frac{1}{d}\boldsymbol{\psi}(\boldsymbol{w})^\top \boldsymbol{w}^*, \quad r_{\psi,d} = \frac{1}{d}\boldsymbol{\psi}(\boldsymbol{w})^\top \boldsymbol{w}, \quad q_{\psi,d} = \frac{1}{d}\|\boldsymbol{\psi}(\boldsymbol{w})\|^2.$$

Assume these converge as $\rho_d \to \rho$, $q_d \to q$, $m_d \to m$, $m_{\psi,d} \to m_\psi$, $r_{\psi,d} \to r_\psi$, $q_{\psi,d} \to q_\psi$. Define the three local fields

$$A_d := \frac{1}{\sqrt{d}}(\boldsymbol{w}^*)^\top \boldsymbol{x}, \quad B_d := \frac{1}{\sqrt{d}}(\boldsymbol{w}^*)^\top \boldsymbol{\psi}(\boldsymbol{x}), \quad C_d := \frac{1}{\sqrt{d}}\boldsymbol{w}^\top \boldsymbol{\psi}(\boldsymbol{x}), \quad D_d := \frac{1}{\sqrt{d}}\boldsymbol{\psi}(\boldsymbol{w})^\top \boldsymbol{\psi}(\boldsymbol{x}).$$

*Proposition* B.2. Conditioning on $\boldsymbol{w}^*$, $\boldsymbol{w}$,

$$(A_d, B_d, C_d, D_d) \underset{d\to\infty}{\Longrightarrow} \mathcal{N}(\mathbf{0}_4, \Sigma),$$

where

$$\Sigma = \begin{pmatrix} \rho & \kappa_\psi \rho & \kappa_\psi m & \kappa_\psi m_\psi \\ \kappa_\psi \rho & \sigma_\psi^2 \rho & \sigma_\psi^2 m & \sigma_\psi^2 m_\psi \\ \kappa_\psi m & \sigma_\psi^2 m & \sigma_\psi^2 q & \sigma_\psi^2 r_\psi \\ \kappa_\psi m_\psi & \sigma_\psi^2 m_\psi & \sigma_\psi^2 r_\psi & \sigma_\psi^2 q_\psi \end{pmatrix}$$

*Proof.* Define $S_d := (A_d, B_d, C_d, D_d)^\top$. For each coordinate $i$, define the following vector:

$$\boldsymbol{\zeta}_{i,d} := \frac{1}{\sqrt{d}}\begin{pmatrix} w_i^{*,(d)} x_i \\ w_i^* \psi(x_i) \\ w_i \psi(x_i) \\ \psi(w_i)\psi(x_i))^\top \end{pmatrix} \in \mathbb{R}^4.$$

Conditioning on $\boldsymbol{w}^*$ and $\boldsymbol{w}$, the vector $\boldsymbol{\zeta}_{i,d}$ are independent with $\mathbb{E}[\boldsymbol{\zeta}_{i,d}] = \mathbf{0}_4$, and $\boldsymbol{S}_d = \sum_{i=1}^{d} \boldsymbol{\zeta}_{i,d}$. Using $\mathbb{E}[\psi(X)] = 0$, $\mathbb{E}[\psi(X)^2] = \sigma_\psi^2$, and $\mathbb{E}[X\psi(X)] = \kappa_\psi$, a direct computation gives

$$\mathrm{Cov}[\boldsymbol{\zeta}_{i,d}] = \frac{1}{d}\begin{pmatrix} (w_i^*)^2 & \kappa_\psi w_i^* & \kappa_\psi w_i^* w_i & \kappa_\psi w_i^* \psi(w_i) \\ \kappa_\psi(w_i^*)^2 & \sigma_\psi^2 (w_i^*)^2 & \sigma_\psi^2 w_i^* w_i & \sigma_\psi^2 w_i^* \psi(w_i) \\ \kappa_\psi w_i^* w_i & \sigma_\psi^2 (w_i^*)^2 & \sigma_\psi^2 w_i^2 & \sigma_\psi^2 w_i \psi(w_i) \\ \kappa_\psi w_i^* \psi(w_i) & \sigma_\psi^2 w_i^* \psi(w_i) & \sigma_\psi^2 w_i \psi(w_i) & \sigma_\psi^2 \psi(w_i)^2 \end{pmatrix}$$

Summing over $i$ yields $\Sigma_d := \mathrm{Cov}[\boldsymbol{S}_d]$ with entries

$$\Sigma_d = \begin{pmatrix} \rho_d & \kappa_\psi \rho_d & \kappa_\psi m_d & \kappa_\psi m_{\psi,d} \\ \kappa_\psi \rho_d & \sigma_\psi^2 \rho_d & \sigma_\psi^2 m_d & \sigma_\psi^2 m_{\psi,d} \\ \kappa_\psi m_d & \sigma_\psi^2 m_d & \sigma_\psi^2 q_d & \sigma_\psi^2 r_{\psi,d} \\ \kappa_\psi m_{\psi,d} & \sigma_\psi^2 m_{\psi,d} & \sigma_\psi^2 r_{\psi,d} & \sigma_\psi^2 q_{\psi,d} \end{pmatrix} \underset{d\to\infty}{\Longrightarrow} \Sigma.$$

Let $\varphi_d(\boldsymbol{u}) = \mathbb{E}[\exp(i\boldsymbol{u}^\top \boldsymbol{S}_d)]$ be the vector characteristic function for $\boldsymbol{u} \in \mathbb{R}^4$. By independence across $i$,

$$\varphi_d(\boldsymbol{u}) = \prod_{i=1}^d \varphi_{i,d}(\boldsymbol{u}), \quad \varphi_{i,d}(\boldsymbol{u}) = \mathbb{E}\Big[e^{i\boldsymbol{u}^\top \boldsymbol{\zeta}_{i,d}}\Big].$$

For a mean-zero real $Y$, $\mathbb{E}[e^{iY}] = 1 - \mathbb{E}[Y^2]/2 + R$ with $|R| \le \mathbb{E}[|Y|^3]/6$. Applying this to $Y = \boldsymbol{u}^\top \boldsymbol{\zeta}_{i,d}$,

$$\varphi_{i,d}(\boldsymbol{u}) = 1 - \frac{1}{2}\boldsymbol{u}^\top \mathrm{Cov}[\boldsymbol{\zeta}_{i,d}]\boldsymbol{u} + r_{i,d}(\boldsymbol{u}), \quad |r_{i,d}(\boldsymbol{u})| \le \frac{1}{6}\mathbb{E}\big[|\boldsymbol{u}^\top \boldsymbol{\zeta}_{i,d}|^3\big].$$

Using $|\psi(\cdot)| \le \omega$ and $(a_1 + \cdots + a_4)^3 \le 64\sum_j a_j^3$,

$$|\boldsymbol{u}^\top \boldsymbol{\zeta}_{i,d}| \le \frac{|u_1||w_i^*|}{\sqrt{d}}|x_i| + \frac{|u_2||w_i^*| + |u_3||w_i| + |u_4||\psi(w_i)|}{\sqrt{d}}|\psi(x_i)|$$

Therefore $\sum_i \mathbb{E}[|\boldsymbol{u}^\top \boldsymbol{\zeta}_{i,d}|^3] = \mathcal{O}_d(d^{-1/2}) \to 0$. Hence

$$\sum_{i=1}^d |r_{i,d}(\boldsymbol{u})| \to 0,$$

Then

$$\log \varphi_d(\boldsymbol{u}) = \sum_{i=1}^d \left(-\frac{1}{2}\boldsymbol{u}^\top \mathrm{Cov}(\boldsymbol{\zeta}_{i,d})\boldsymbol{u}\right) + o_d(1) = -\frac{1}{2}\boldsymbol{u}^\top \Sigma_d \boldsymbol{u}.$$

Since $\Sigma_d \underset{d\to\infty}{\Longrightarrow} \Sigma$,

$$\varphi_d(\boldsymbol{u}) \underset{d\to\infty}{\Longrightarrow} e^{-\frac{1}{2}\boldsymbol{u}^\top \Sigma \boldsymbol{u}}, \quad \forall \boldsymbol{u} \in \mathbb{R}^4.$$

The limit is the characteristic function of $\mathcal{N}(\boldsymbol{0}_d, \Sigma)$ and is continuous at $\boldsymbol{u} = \boldsymbol{0}_4$. By Lévy's continuity theorem, we conclude

$$\boldsymbol{S}_d = (A_d, B_d, C_d, D_d)^\top \underset{d}{\Longrightarrow} \mathcal{N}(\boldsymbol{0}_4, \Sigma_d).$$

$\square$

## B.3. Generalization Error

We express the mean-squared prediction error on a fresh sample $(\boldsymbol{x}_{\mathrm{new}}, y_{\mathrm{new}})$ drawn from the same distribution, when the model predicts with the quantized weights and quantized inputs.

*Proposition* B.3.

$$\varepsilon_g \underset{d\to\infty}{\Longrightarrow} \sigma^2 + \rho + \sigma_\psi^2 q_\psi - 2\kappa_\psi m_\psi.$$

*Proof.*

$$\varepsilon_g := \mathbb{E}[(y - \hat{y})^2]$$

$$= \sigma^2 + \mathbb{E}\left[\left(\frac{1}{\sqrt{d}}\boldsymbol{x}^\top \boldsymbol{w}^*\right)^2\right] + \mathbb{E}\left[\left(\frac{1}{\sqrt{d}}\boldsymbol{\psi}(\boldsymbol{w})^\top \boldsymbol{\psi}(\boldsymbol{x})\right)^2\right] - 2\mathbb{E}\left[\frac{1}{\sqrt{d}}\boldsymbol{x}^\top \boldsymbol{w}^* \frac{1}{\sqrt{d}}\boldsymbol{\psi}(\boldsymbol{w})^\top \boldsymbol{\psi}(\boldsymbol{x})\right]$$

$$= \sigma^2 + \rho_d + \sigma_\psi^2 q_{\psi,d} - 2\kappa_\psi m_{\psi,d}.$$

$\square$

## C. Derivation of Concentration in Microscopic State

In this section, we present a formal approach that enables us to exactly derive the limiting PDE in Eq. (4). This asymptotic characterization has been rigorously established in (Wang et al., 2017). We direct readers to (Wang et al., 2017; 2019) for a comprehensive framework that rigorously establishes the above scaling limit.

We investigate the one-pass STE update

$$\boldsymbol{w}^{t+1} = \boldsymbol{w}^t - \eta\left[\frac{(\hat{y}(\boldsymbol{x}^t;\boldsymbol{w}) - y^t)}{\sqrt{d}}\boldsymbol{\psi}(\boldsymbol{x}^t) + \frac{\lambda}{d}\boldsymbol{\psi}(\boldsymbol{w}^t)\right], \tag{9}$$

with

$$\boldsymbol{x}_\mu \sim \mathcal{N}(\boldsymbol{x}_\mu; \boldsymbol{0}, I_d),\ y_\mu = y(\boldsymbol{x}_\mu; \boldsymbol{w}^*) = \frac{1}{\sqrt{d}}\boldsymbol{x}_\mu^\top\boldsymbol{w}^* + \xi_\mu,\ \hat{y}(\boldsymbol{x};\boldsymbol{w}) = \frac{1}{\sqrt{d}}\boldsymbol{\psi}(\boldsymbol{w})^\top\boldsymbol{\psi}(\boldsymbol{x}).$$

The learning rate is $\eta \in \mathbb{R}_+$; $\xi^t$ is independent noise with $\mathbb{E}[\xi^t] = 0$, $\mathrm{Var}[\xi^t] = \sigma^2$.

In addition, the assumptions used in the following derivations are presented below.

*Assumption* C.1. We define the macroscopic state $\Psi^t = (m^t, q^t) \in \mathbb{R}^2$ by $m^t = \boldsymbol{w}^{*\top}\boldsymbol{w}^t/d$, $q^t = \|\boldsymbol{w}^t\|^2/d$.

(1) The pairs $(\boldsymbol{x}^t, \xi^t)_{t\in[T]}$ are i.i.d. random variables across $t \in [T]$.

(2) The initial macroscopic state $\Psi^0$ satisfies $\mathbb{E}\|\Psi^0 - \bar{\Psi}^0\| \leq C/\sqrt{d}$, where $\bar{\Psi}^0 \in \mathbb{R}^2$ is a deterministic vector and $C$ is a constant independent of $d$.

(3) The fourth moments of the initial parameter $\boldsymbol{w}^0$ are uniformly bounded: $\mathbb{E}\sum_{i=1}^d (w_i^*)^4 + (w_i^0)^4 \leq C$ where $C$ is a constant independent of $d$.

### C.1. Formal Derivation of Limiting PDE

For the $i$-th coordinate define

$$\Delta w_t^i := w_i^{t+1} - w_i^t = -\eta\left[g_i^t + \frac{\lambda}{d}\psi_T(w_i^t)\right]. \tag{10}$$

where

$$g_i^t := \left(\frac{1}{\sqrt{d}}(\boldsymbol{\psi}_T(\boldsymbol{w}^t)^\top\boldsymbol{\psi}_T(\boldsymbol{x}^t) - (\boldsymbol{w}^*)^\top\boldsymbol{x}^t) - \xi^t\right)\frac{\psi_T(x_i^t)}{\sqrt{d}}$$

**One-Step Conditional Moments.** We start with the leading terms of the drift and diffusion of $\Delta w_i^t$.

*Lemma* C.2. Under Assumption C.1, for every $i$ and $t$,

$$\mathbb{E}_t[\Delta w_i^t] = \frac{\eta}{d}[\kappa_\psi w_i^* - (\sigma_\psi^2 + \lambda)\psi_T(w_i^t)], \quad \forall i \in [d], \forall t \in [T].$$

*Proof.* By the independence of $(x_i^t)_{i\in[d]}$ and $\mathbb{E}[\psi(X)] = 0$,

$$\mathbb{E}_t[g_i^t] = \frac{1}{d}\sum_j \mathbb{E}_t\left[(\psi_T(w_j^t)\psi_T(x_j^t) - w_j^* x_j^t)\psi_T(x_i^t)\right]$$

$$= \frac{1}{d}(\psi_T(w_i^t)\mathbb{E}_t[\psi_T(x_i^t)^2] - w_j^*\mathbb{E}_t[x_i^t\psi_T(x_i^t)])$$

$$= \frac{1}{d}(\sigma_\psi^2\psi_T(w_i^t) - \kappa_\psi w_i^*).$$

Substitute in $\Delta w_i^t = -\eta(g_i^t + \lambda\psi(w_i^t)/d)$ to get the claim. $\square$

*Lemma* C.3. Under Assumption C.1, for every $i$ and $t$,

$$\mathbb{E}_t[(\Delta w_i^t)^2] = \frac{\eta^2}{d}\sigma_\psi^2\varepsilon_g(t) + \mathcal{O}_d(d^{-2}), \quad \forall i \in [d], \forall t \in [T].$$

*Proof.* Under Assumption C.1, for every $i$ and $t$,

$$
\begin{aligned}
\mathbb{E}_t\big[(\Delta w_i^t)^2\big] &= \eta^2 \mathbb{E}_t\Big[\big(g_i^t\big)^2\Big] + \frac{\eta^2 \lambda^2}{d^2} \psi_T(w_i^t)^2 + \frac{2\eta^2 \lambda}{d} \psi_T(w_i^t) \mathbb{E}_t\big[g_i^t\big] \\
&= \eta^2 \mathbb{E}_t\Big[\big(g_i^t\big)^2\Big] + \mathcal{O}_d(d^{-2}) \\
&= \frac{1}{d} \sigma_\psi^2 \big(\rho + \sigma^2 + \sigma_\psi^2 q_\psi^t - 2\kappa_\psi m_\psi^t\big) + \mathcal{O}_d(d^{-2}).
\end{aligned}
$$

$\square$

*Lemma* C.4.
$$
\mathbb{E}_t\big[|\Delta w_i^t|^3\big] = \mathcal{O}_d(d^{-3/2}), \quad \forall i \in [d], \forall t \in [T].
$$

*Proof.* From Eq. (10) and the boundness $|\psi| \le \omega$,

$$
\begin{aligned}
|g_i^t|^3 &= \frac{\left| \frac{1}{\sqrt{d}} (\boldsymbol{\psi}_T(\boldsymbol{w}^t)^\top \boldsymbol{\psi}_T(\boldsymbol{x}^t) - (\boldsymbol{w}^*)^\top \boldsymbol{x}^t) - \xi^t \right|^3 |\psi(x_i^t)|^3}{d^{3/2}} \\
&\le \frac{\omega^3}{d^{3/2}} \left| \frac{1}{\sqrt{d}} (\boldsymbol{\psi}_T(\boldsymbol{w}^t)^\top \boldsymbol{\psi}_T(\boldsymbol{x}^t) - (\boldsymbol{w}^*)^\top \boldsymbol{x}^t) - \xi^t \right|^3 \\
&\le \frac{\omega^3}{d^{3/2}} \left( \frac{1}{\sqrt{d}} \|\boldsymbol{\psi}(\boldsymbol{w}^t)\| \|\boldsymbol{\psi}(\boldsymbol{x}^t)\| + \frac{1}{\sqrt{d}} \|\boldsymbol{w}^*\| \|\boldsymbol{x}^t\| + |\xi^t| \right)^3.
\end{aligned}
$$

Using $(a+b+c)^3 \le 27(a^3 + b^3 + c^3)$ and Lemma D.6,

$$
|g_i^t|^3 \le \frac{C\omega^3}{d^{3/2}} = \mathcal{O}_d(d^{-3/2}).
$$

Finally,

$$
|\Delta w_i^t|^3 \le C(|g_i^t|^3 + d^{-3}) = \mathcal{O}_d(d^{-3/2}).
$$

$\square$

The three Lemmas identify the drift and diffusion to leading order and show that the characteristic time of the process is $1/d$.

**Decomposition on Test Functions.** The convergence of empirical measures can be studied through their actions on test functions. Let $\zeta(w^*, w)$ be a nonnegative, bounded and $C^3$ test function. From the update equation Eq. (9), a third-order Taylor expansion at $w_i^t$ yields

$$
\mathbb{E}_{\mu_{t+1}^{(d)}}[\zeta] - \mathbb{E}_{\mu_t^{(d)}}[\zeta] = \frac{1}{d} \sum_{i=1}^d \partial_w \zeta(w_i^t, w_i^*) \Delta w_i^t + \frac{1}{2d} \sum_{i=1}^d \partial_w^2 \zeta(w_i^t, w_i^*)(\Delta w_i^t)^2 + r_t,
$$

with the following remainder

$$
r_t := \frac{1}{6d} \sum_{i=1}^d \partial_w^3 \zeta(c_i^t, w_i^*)(\Delta w_i^t)^3, \quad c_i^t \in [w_i^t, w_i^{t+1}].
$$

Introducing two sequences

$$
\begin{aligned}
v_t &:= \mathbb{E}_t\Big[ \mathbb{E}_{\mu_{t+1}^{(d)}}[\zeta] - \mathbb{E}_{\mu_t^{(d)}}[\zeta] - r_k \Big], \\
m_t &:= \mathbb{E}_{\mu_{t+1}^{(d)}}[f] - \mathbb{E}_{\mu_t^{(d)}}[f] - r_k - v_k,
\end{aligned}
$$

satisfying

$$
\mathbb{E}_{\mu_{t+1}^{(d)}}[f] - \mathbb{E}_{\mu_t^{(d)}}[f] = v_t + m_t + r_t.
$$

Summing over $t \in [T]$ gives

$$\mathbb{E}_{\mu_{t+1}^{(d)}}[f] - \mathbb{E}_{\mu_0^{(d)}}[f] = \sum_{l<t} v_l + \sum_{l<t} m_l + \sum_{l<t} r_l = V_t + M_t + R_t,$$

where

$$V_t := \sum_{l<t} v_l, \ M_t := \sum_{l<t} m_l, \ R_t := \sum_{l<t} r_l.$$

We also set $V_0 = M_0 = R_0$. From Lemma C.2 and C.3,

$$v_t = \frac{1}{d}\mathbb{E}_{\mu_t^{(d)}}\left[\eta\big(\kappa_\psi w^* - (\sigma_\psi^2 + \lambda)\psi(w)\big)\,\partial_w\zeta\right] + \frac{\eta^2\sigma_\psi^2}{2d}\varepsilon_g(t)\mathbb{E}_{\mu_t^{(d)}}[\partial^2\zeta] + \mathcal{O}_d(d^{-3/2}). \tag{11}$$

Embedding the discrete sequence $V_t$ in continuous-time by factor of $d$, we define

$$V(\tau) := V_{\lfloor d\times\tau\rfloor},$$

Similarly, we can define $M(\tau)$, $R(\tau)$, $\mu_\tau$ as the continuous-time rescaled versions of their discrete-time counterparts. Since $V(\tau)$ is piecewise-constant over intervals of length $1/d$, the expression in Eq. (11) can be written as

$$v_t = \int_{t/d}^{t+1/d} L(\mu_s^{(d)})ds + \mathcal{O}(d^{-1/2}),$$

where

$$L(\mu_s) := \mathbb{E}_{\mu_t^{(d)}}\left[\eta\big(\kappa_\psi w^* - (\sigma_\psi^2 + \lambda)\psi(w)\big)\,\partial_w\zeta\right] + \frac{\eta^2\sigma_\psi^2}{2}\varepsilon_g(t)\mathbb{E}_{\mu_t^{(d)}}[\partial_w^2\zeta].$$

It follows that

$$\mathbb{E}_{\mu_{t+1}^{(d)}}[\zeta] - \mathbb{E}_{\mu_0^{(d)}}[\zeta] = \int_0^t L(\mu_s^{(d)})ds + M_t + R_t.$$

We note that $L(\mu_s^{(d)})$ contains exactly the last two terms on the right-hand side of the limiting PDE in Eq. (4). Formally, if the martingale term $M(\tau)$ and the higher-order term $R(\tau)$ converge to 0 as $d \to +\infty$, we can then establish the scaling limit. We refer readers to (Wang et al., 2017; 2019) for a rigorous proof, following a standard recipe in the literature (Méléard & Roelly-Coppoletta, 1987) and (Sznitman, 2006)

## D. Proof of Concentration on ODE

In this section, we prove Theorem 5.3 in the main text based on the following two Lemmas. (I) The first moment of the increment of the macroscopic stochastic process $\Psi^t$ converges. (II) The second moment of the increment vanishes. Intuitively, these conditions imply that the leading-order behavior of the average increment is governed by the ODEs in Theorem 5.3, and that as the input dimension grows, the stochastic component of the increment vanishes.

The proof is organized as follows. First, we establish the two conditions in the next subsection. Second, we demonstrate that these conditions are sufficient to prove Theorem 5.3. Finally, we compile the technical lemmas that are applied repeatedly in the preceding arguments. Our approach adheres to standard convergence arguments for stochastic processes (Kushner & Yin, 2009; Billingsley, 2013; Wang et al., 2018).

### D.1. Proof of Isotropy

We begin by stating the isotropy reduction that expresses $m_\psi$, $q_\psi$, and $r_\psi$ in terms of the macroscopic state $m$, $q$ via $s = \sqrt{q - m^2/\rho}$.

*Proposition* D.1. Let $s^t = (q^t - (m^t)^2/\rho)^{1/2}$. Under Assumption 5.1, for any $t \in [T]$,

$$m_\psi(m^t, s^t) = \mathbb{E}_{w^*}\left[w^*\left(-\omega + \Delta \sum_{i=1}^{L} \Phi\left(\frac{\frac{m^t w^*}{\rho} - \theta_i}{s^t}\right)\right)\right],$$

$$q_\psi(m^t, s^t) = \mathbb{E}_{w^*}\left[v_0^2 + \sum_{i=1}^{L}(v_i^2 - v_{i-1}^2)\Phi\left(\frac{\frac{m^t w^*}{\rho} - \theta_i}{s^t}\right)\right],$$

$$r_\psi(m^t, s^t) = \frac{m^t m_\psi}{\rho} + \Delta s^t \sum_{i=1}^{L} \mathbb{E}_{w^*}\left[\phi\left(\frac{\frac{m^t w^*}{\rho} - \theta_i}{s^t}\right)\right].$$

*Proof.* By Assumption 5.1, each coordinate $i$ is expressed as follows:

$$w_i^t = \frac{m^t}{\rho}w_i^* + s^t \xi_i, \quad \xi_i \sim_{\text{i.i.d}} \mathcal{N}(0, 1),$$

with $\{\xi_i\}$ independent of $\{w_i^*\}$. For any bounded measurable $f : \mathbb{R}^2 \to \mathbb{R}$, Assumption 5.1 implies

$$\frac{1}{d}\sum_{i=1}^{d} f(w_i^*, w_i^t) \underset{\text{a.s.}}{\Longrightarrow} \mathbb{E}_{w^*}\mathbb{E}_z[f(w^*, z)], \quad z = \frac{m^t}{\rho}w^* + s^t\xi. \tag{12}$$

We apply Eq. (12) to the choice $f(u, v) \in \{u\psi_T(v), \psi_T(v), v\psi_T(v)\}$, given by

$$m_\psi(m^t, s^t) = \mathbb{E}_{w^*}\mathbb{E}_z[w^*\psi_T(z)],$$

$$q_\psi(m^t, s^t) = \mathbb{E}_{w^*}\mathbb{E}_z\left[\psi_T(z)^2\right],$$

$$r_\psi(m^t, s^t) = \mathbb{E}_{w^*}\mathbb{E}_z[z\psi_T(z)],$$

Then, taking the limit $T \to +0$ proves the lemma. First,

$$\begin{aligned}
m_\psi(m^t, s^t) &= \mathbb{E}_{w^*}\left[\mathbb{E}_\xi\left[w^*\psi_T\left(\frac{m^t}{\rho}w^* + s^t\xi\right)\right]\right] \\
&= \mathbb{E}_{w^*}\left[w^*\left(-\omega + \Delta \sum_{k=1}^{L}\mathbb{E}_z\left[\Phi\left(\frac{z - \theta_k}{T}\right)\right]\right)\right] \\
&= \mathbb{E}_{w^*}w^*\mathbb{E}_z\left(-\omega + \Delta \sum_{k=1}^{L}\mathbb{E}_\xi\left[\Phi\left(\frac{z - \theta_k}{T}\right)\right]\right) \\
&= \mathbb{E}_{w^*}\left[w^*\left(-\omega + \Delta \sum_{k=1}^{L}\Phi\left(\frac{mw^*/\rho - \theta_k}{\sqrt{(s^t)^2 + T^2}}\right)\right)\right]
\end{aligned}$$

In the last step, we use Lemma A.1.

$$\begin{aligned}
\mathbb{E}[z\psi_T(z)] &= \mathbb{E}_{w^*}\mathbb{E}_z\left[z\left(-\omega + \Delta \sum_{k=1}^{L}\mathbb{E}_z\left[\Phi\left(\frac{z - \theta_k}{T}\right)\right]\right)\right] \\
&= \mathbb{E}_{w^*}\left[-\frac{mw^*}{\rho}\omega + \Delta \sum_{k=1}^{L}\mathbb{E}_z\left[z\Phi\left(\frac{z - \theta_k}{T}\right)\right]\right] \\
&= \frac{m^t}{\rho}m_\psi(m^t, s^t) + \Delta\frac{(s^t)^2}{\sqrt{(s^t)^2 + T^2}}\sum_{k=1}^{L}\mathbb{E}_{w^*}\left[\phi\left(\frac{m^t w^*/\rho - \theta_k}{\sqrt{(s^t)^2 + T^2}}\right)\right].
\end{aligned}$$

The last equality follows from Lemma A.4. $q_\psi(m^t, s^t)$ is also expressed as follows:

$$
\mathbb{E}_{w^*}\mathbb{E}_z\big[\psi_T(z)^2\big] = \omega^2 - 2\omega\Delta\sum_{k=1}^{L}\mathbb{E}_{w^*}\mathbb{E}_z\left[\Phi\left(\frac{z-\theta_k}{T}\right)\right] + \Delta^2\sum_{ij}\mathbb{E}_{w^*}\mathbb{E}_z\left[\Phi\left(\frac{z-\theta_i}{T}\right)\Phi\left(\frac{z-\theta_j}{T}\right)\right]
$$

$$
= \mathbb{E}_{w^*}\left[\omega^2 - 2\omega\Delta\sum_{k}\Phi\left(\frac{mw^*/\rho - \theta_k}{\sqrt{(s^t)^2 + T^2}}\right)\right]
$$

$$
+ \mathbb{E}_{w^*}\left[\Delta^2\sum_{ij}\mathbb{E}_{w^*}\left[\Phi_2\left(\frac{m^tw^*/\rho - \theta_i}{\sqrt{(s^t)^2 + T^2}}, \frac{m^tw^*/\rho - \theta_j}{\sqrt{(s^t)^2 + T^2}}; \frac{(s^t)^2}{(s^t)^2 + T^2}\right)\right]\right],
$$

As $T \to +0$, we have

$$
\sqrt{(s^t)^2 + T^2} \to s^t, \quad \frac{(s^t)^2}{(s^t)^2 + T^2} \to 1,
$$

thus,

$$
\Phi\left(\max\left\{\frac{m^tw^*/\rho - \theta_i}{s^t}, \frac{m^tw^*/\rho - \theta_j}{s^t}\right\}\right),
$$

which completes the proof. □

## D.2. Convergence of First Moments of Increment to ODEs

We first review the training algorithm of STE which characterizes a Markov process $w^t$. The specific update rule is given by

$$
w^{t+1} = w^t - \eta\left[\left(\frac{1}{\sqrt{d}}\psi(w)^\top\psi(x) - \frac{1}{\sqrt{d}}x_\mu^\top w^* - \xi_\mu\right)\frac{1}{\sqrt{d}}\psi(x^t) + \frac{\lambda}{d}\psi(w^t)\right],
$$

The following lemma holds for the macroscopic state $\Psi^t$ characterized by the above updates.

*Lemma* D.2. Under Assumptions C.1, for all $t < dT$ the following inequality holds:

$$
\mathbb{E}\left\|\mathbb{E}_t\Psi^{t+1} - \Psi^t - \frac{1}{d}F(\Psi^t)\right\| \leq \frac{C}{d^{3/2}}. \tag{13}
$$

*Proof.* Recall that $\Psi^t = (m^t, q^t) \in \mathbb{R}^2$ is composed of two scalar. $\|\Psi^t\| \leq |\Psi^t|$ holds. Thus, the following inequality is sufficient to prove Eq. (13):

$$
\mathbb{E}\left|\mathbb{E}_t\Psi_i^{t+1} - \Psi_i^t - \frac{1}{d}F_i(\Psi)\right| \leq \frac{C}{d^{3/2}}, \tag{14}
$$

where $\Psi_i^t$ is $i$ element of $\Psi^t$. Subsequently, we show that the above inequality holds for each element of $\Psi^t$.

For $m^t$, the following stronger result is obtained:

$$
\mathbb{E}_t m^{t+1} - m^t + \frac{\eta}{d}F_1(\Psi^t) = 0, \tag{15}
$$

where $F_1(\Psi)$ is defined in Eq. 25. This is directly proved by multiplying $(w^*)^\top/d$ from the left on both sides of Eq. D.2, which yields

$$
m^{t+1} = m^t - \frac{\eta}{d}\big[(D^t - A^t - \xi_\mu)B^t + m_\psi^t\lambda\big],
$$

Then, taking the conditional expectation $\mathbb{E}_t$ on both sides and using Lemma B.2, we reach Eq. 15:

$$
\mathbb{E}_t m^{t+1} = m^t - \frac{\eta}{d}\mathbb{E}_t\big[(D^t - A^t - \xi_\mu)B^t + m_\psi^t\lambda\big],
$$

$$
= m^t - \frac{\eta}{d}\big(\sigma_\psi^2 m^t - \kappa_\psi\rho + m_\psi^t\lambda\big).
$$

Next, for $q^t$, the following inequality holds:

$$
\mathbb{E}_t q^{t+1} - q^t - \frac{1}{d}F_2(\Psi^t) \leq \frac{C}{d^{\frac{3}{2}}}, \tag{16}
$$

where $F_2$ is defined in Eq. (26). This is proved by evaluating $q^t = (\boldsymbol{w}^{t+1})^\top \boldsymbol{w}^{t+1}/d$ as follows:

$$
\begin{aligned}
q^{t+1} &= \frac{1}{d}(\boldsymbol{w}^{t+1})^\top \boldsymbol{w}^{t+1} \\
&= \frac{1}{d}\left(\boldsymbol{w}^t - \eta\left[\boldsymbol{g}^t + \frac{\lambda}{d}\psi(\boldsymbol{w})\right]\right)^\top \left(\boldsymbol{w}^t - \eta\left[\boldsymbol{g}^t + \frac{\lambda}{d}\psi(\boldsymbol{w})\right]\right) \\
&= q^t - \frac{2\eta}{d}\boldsymbol{w}^\top\left[\boldsymbol{g}^t + \frac{\lambda}{d}\psi(\boldsymbol{w})\right] + \frac{\eta^2}{d}\left\|\boldsymbol{g}^t + \frac{\lambda}{d}\psi(\boldsymbol{w})\right\|^2.
\end{aligned}
$$

Then taking the conditional expectation $\mathbb{E}_t$ and using Proposition B.2 and Lemma D.6,

$$
\begin{aligned}
\mathbb{E}_t q^{t+1} &= q^t - \frac{2\eta}{d}\mathbb{E}_t\boldsymbol{w}^\top\boldsymbol{g}^t - 2\eta\lambda r_\psi^t + \mathbb{E}_t\frac{\eta^2}{d}\|\boldsymbol{g}^t\|^2 + \mathcal{O}(d^{-2}) \\
&= q^t - \frac{2\eta}{d}\mathbb{E}_t\big((D^t - A^t)C^t\big) - 2\eta\lambda m^t + \frac{\eta^2}{d}\mathbb{E}_t\left\|(D^t - A^t - \xi_t)\frac{1}{\sqrt{d}}\psi(\boldsymbol{x}^t)\right\|^2 + \mathcal{O}(d^{-2}) \\
&= q^t - 2\eta\big((\sigma_\psi^2 + \lambda)r_\psi^t - \kappa_\psi m_t\big) + \eta^2\sigma_\psi^2\varepsilon_g^t + \mathcal{O}(d^{-2}).
\end{aligned}
$$

Combining Eq. (15) and Eq. (16), Eq. (14) is proven, which concludes the whole proof. $\qquad\square$

### D.3. Convergence of Second Moments of Increments

We now proceed to bound the second-order moments of the increments.

*Lemma* D.3. Under Assumption C.1, for all $t < d \times T$ the following inequality holds:

$$
\mathbb{E}\|\Psi^{t+1} - \mathbb{E}_t\Psi^{t+1}\|^2 \le \frac{C}{d^2}.
$$

*Proof.* Note that

$$
\begin{aligned}
\mathbb{E}\|\Psi^{t+1} - \mathbb{E}_t\Psi^{t+1}\|^2 &= \mathbb{E}\|\Psi^{t+1} - \Psi^t - \mathbb{E}_t(\Psi^{t+1} - \Psi^t)\|^2, \\
&\le \mathbb{E}\|\Psi^{t+1} - \Psi^t\|^2 + \mathbb{E}\|\mathbb{E}_t\Psi^{t+1} - \Psi^t\|^2, \\
&\le \mathbb{E}\|\Psi^{t+1} - \Psi^t\|^2 + \mathbb{E}\left\|\frac{1}{d}F(\Psi^t) + \frac{C}{d^{\frac{3}{2}}}\right\|^2, \\
&\le \mathbb{E}\|\Psi^{t+1} - \Psi^t\|^2 + \frac{C}{d^2}.
\end{aligned}
$$

Here the third line is due to Lemma D.2. Thus, it is sufficient to prove that

$$
\mathbb{E}\|\Psi^{t+1} - \Psi^t\|^2 \le \frac{C}{d^2}. \tag{17}
$$

In the following, the second moment of each element in $\Psi^{t+1} - \Psi^t$ will be bounded. For $m^t$, Proposition B.2 and Lemma D.6,

$$
\mathbb{E}(m^{t+1} - m^t)^2 = \frac{\eta^2}{d^2}\mathbb{E}\big((D^t - A^t - \xi_\mu)B^t + m_\psi^t\lambda\big)^2 = \mathcal{O}_d(d^{-2}). \tag{18}
$$

For $q^t$, Proposition B.2 and Lemma D.6,

$$
\mathbb{E}(q^{t+1} - q^t)^2 = \mathbb{E}\left(-\frac{2\eta}{d}\boldsymbol{w}^\top\left[\boldsymbol{g}^t + \frac{\lambda}{d}\psi(\boldsymbol{w})\right] + \frac{\eta^2}{d}\left\|\boldsymbol{g}^t + \frac{\lambda}{d}\psi(\boldsymbol{w})\right\|^2\right)^2 \tag{19}
$$

$$
= \mathbb{E}\left(-\frac{2\eta}{d}\big[\boldsymbol{w}^\top\boldsymbol{g}^t + r_\psi^t\big] + \frac{\eta^2}{d}\|\boldsymbol{g}^t\|^2\right)^2 + \mathcal{O}_d(d^{-2}) = \mathcal{O}_d(d^{-2})
$$

Combining these results Eq. (18) and Eq. (19), Eq. (17) is proven, which concludes the whole proof. $\qquad\square$

### D.4. Proof of Theorem 5.3

In this section, we complete the remaining proof of Theorem 5.3 from Lemmas D.2 and D.3 by employing a coupling argument, following the approach of (Ichikawa & Hukushima, 2024; Wang et al., 2019).

*Proof.* The proof uses the coupling trick. In particular, we first define a stochastic process $\mathcal{B}^t$ that is coupled with the process $\Psi^t$ as

$$\mathcal{B}^{t+1} = \mathcal{B}^t + \frac{1}{d}F(\mathcal{B}^t) + \Psi^{t+1} - \mathbb{E}_t \Psi^{t+1}$$

with the deterministic initial condition $\mathcal{B}^0 = \bar{\Psi}^0$. For this stochastic process $\mathcal{B}^t$, the following inequality holds for all $t \leq d \times T$:

$$\mathbb{E}\|\mathcal{B}^t - \Psi^t\| \leq \frac{C}{d^{1/2}}. \tag{20}$$

This inequality is proved as follows.

$$\mathbb{E}\|\mathcal{B}^{t+1} - \Psi^{t+1}\| \leq \mathbb{E}\|\mathcal{B}^t - \Psi^t\| + \frac{1}{d}\mathbb{E}\|F(\mathcal{B}^t) - F(\Psi^t)\| + \mathbb{E}\left\|\mathbb{E}_t\Psi^{t+1} - \Psi^t - \frac{1}{d}F(\Psi^t)\right\|.$$

From Lemma D.2 and Lemma **??** in subsequent Sec. D.5, one can get

$$\mathbb{E}\|\mathcal{B}^{t+1} - \Psi^{t+1}\| \leq \mathbb{E}\|\mathcal{B}^t - \Psi^t\| + L\|\mathcal{B}^t - \Psi^t\| + Cd^{-\frac{3}{2}}$$
$$\leq (1 + Ld^{-1})\|\mathcal{B}^t - \Psi^t\| + Cd^{-\frac{3}{2}}.$$

Applying this bound iteratively, for all $t \leq dT$, one can expand as follows:

$$\mathbb{E}\|\mathcal{B}^t - \Psi^t\| \leq e^{LT}\left(\mathbb{E}\|\mathcal{B}^0 - \Psi^0\| + \frac{C}{L}d^{-\frac{1}{2}}\right) \leq \frac{C}{d^{\frac{1}{2}}}.$$

For the last inequality, we use Assumption C.1 in the main text.

Next, we define a deterministic process $\mathcal{S}^t$ as follows:

$$\mathcal{S}^{t+1} = \mathcal{S}^t + \frac{1}{d}F(\mathcal{S}^t)$$

with the deterministic initial condition $\mathcal{S}^0 = \bar{\Psi}^0$. Similarly, the following inequality holds for all $t \leq d \times T$:

$$\mathbb{E}\|\mathcal{B}^t - \mathcal{S}^t\|^2 \leq \frac{C}{d} \tag{21}$$

To prove this inequality, one can express as

$$\mathbb{E}\|\mathcal{B}^{t+1} - \mathcal{S}^{t+1}\|^2 = \mathbb{E}\|\mathcal{B}^t - \mathcal{S}^t\|^2 + \frac{1}{d^2}\mathbb{E}\|F(\mathcal{B}^t) - F(\mathcal{S}^t)\|^2 + \frac{2}{d}\mathbb{E}(F(\mathcal{B}^t) - F(\mathcal{S}^t))^\top(\mathcal{B}^t - \mathcal{S}^t) + \mathbb{E}\|\Psi^{t+1} - \mathbb{E}_t\Psi^{t+1}\|^2.$$

Here, one uses the identity given by

$$\mathbb{E}_t(\Psi^{t+1} - \mathbb{E}_t\Psi^{t+1})^\top(\mathcal{B}^t - \mathcal{S}^t) = \mathbb{E}_t(\Psi^{t+1} - \mathbb{E}_t\Psi^{t+1})^\top(F(\mathcal{B}^t) - F(\mathcal{S}^t)) = 0.$$

Then, from Lemma D.3, one can get following inequality:

$$\mathbb{E}\|\mathcal{B}^{t+1} - \mathcal{S}^{t+1}\|^2 \leq \left(1 + \frac{CL}{d}\right)\mathbb{E}\|\mathcal{B}^t - \mathcal{S}^t\|^2 + \frac{C}{d^2}.$$

Applying this bound iteratively, for all $t \leq d \times T$, Eq. (21) is proven as follows:

$$\mathbb{E}\|\mathcal{B}^t - \mathcal{S}^t\|^2 \leq \frac{C}{d}.$$

Note that $\mathcal{S}^t$ is a standard first-order finite difference approximation of the ODEs with the step size $1/d$. The standard Euler argument implies that

$$\|\mathcal{S}^t - \Psi(t)\| \leq \frac{C}{d}. \tag{22}$$

Finally, combining Eq. (20), (21) and (22), Theorem 5.3 is proven as follows:

$$\begin{aligned}
\mathbb{E}\|\Psi^t - \Psi(t)\| = \mathbb{E}\|\Psi^t - \mathcal{B}^t + \mathcal{B}^t - \mathcal{S}^t + \mathcal{S}^t - \Psi(t)\| \\
\leq \mathbb{E}\|\Psi^t - \mathcal{B}^t\| + \mathbb{E}\|\mathcal{B}^t - \mathcal{S}^t\| + \mathbb{E}\|\mathcal{S}^t - \Psi(t)\| \\
\leq \mathbb{E}\|\Psi^t - \mathcal{B}^t\| + (\mathbb{E}\|\mathcal{B}^t - \mathcal{S}^t\|^2)^{\frac{1}{2}} + \mathbb{E}\|\mathcal{S}^t - \Psi(t)\| \\
\leq \frac{C}{d^{\frac{1}{2}}}.
\end{aligned}$$

$\square$

### D.5. Extra Proofs

In this section we collect several auxiliary bounds that are repeatedly used in the proof of Theorem 5.3.

#### D.5.1. BOUND FOR MICROSCOPIC STATE

*Lemma D.4.* Under the same assumption as in Theorem 5.3, there exists a constant $C$ such that, for any integer $l \in \{2, 3, 4\}$ and any $i \in [d]$,

$$\mathbb{E}_t\left[|\Delta w_i^t|^l\right] \leq \frac{C}{d^{l/2}}\left(1 + \frac{1}{d}\sum_j |w_j^t|^2\right).$$

*Proof.* By the triangle inequality and Minkowski's inequality, for some $C > 0$,

$$\mathbb{E}_t|\Delta w_i^t|^l \leq C\eta^l\left(\mathbb{E}_t|g_i^t|^l + \left|\frac{\lambda}{d}\psi(w_i^t)\right|^l\right).$$

For $l = 2, 3, 4$, we interpolate between second and fourth moments via Hölder interpolation:

$$\mathbb{E}_t[|g_i^t|^l] \leq \left(\mathbb{E}_t(g_i^t)^2\right)^{\frac{4-l}{2}}\left(\mathbb{E}_t(g_i^t)^4\right)^{\frac{l-2}{2}}. \tag{23}$$

By independence and the Wick's formula, there exist constants $C, C'$ independent of $d$ such that

$$\mathbb{E}[(g_i^t)^2] \leq \frac{C}{d}\left(\frac{1}{d}\|\psi(\boldsymbol{w}^t)\|^2 + \rho^2 + \sigma^2\right), \quad \mathbb{E}[(g_i^t)^4] \leq \frac{C'}{d^2}\left(\frac{1}{d}\|\psi(\boldsymbol{w}^t)\|^2 + \rho^2 + \sigma^2\right)^2, \tag{24}$$

Plugging (24) into (23) yields, for $l \in \{2, 3, 4\}$,

$$\mathbb{E}_t[|g_i^t|^l] \leq \frac{C}{d^{l/2}}\left(\frac{1}{d}\sum_i |\psi(w_i)|^l + \rho + \sigma^2\right)^{l/2}.$$

Thus,

$$\begin{aligned}
\mathbb{E}_t\left[|\Delta w_i^t|^l\right] \leq C\left[\frac{1}{d^{l/2}}\left(1 + \frac{1}{d}\sum_j |w_j^t|^l\right) + \frac{1}{d^4}\sum_i \psi(w_i^l)^l\right] \\
\leq \frac{C}{d^{l/2}}\left(1 + \frac{1}{d}\sum_j |w_j^t|^l\right).
\end{aligned}$$

$\square$

*Lemma* D.5. Under the same assumption as in Theorem 5.3, there exists a constant $C$ such that

$$\max_{0 \le t \le Td} \sum_{i=1}^{d} \mathbb{E}(w_i^t)^4 \le C.$$

*Proof.* By the binomial expansion,

$$\mathbb{E}(w_i^{t+1})^4 - \mathbb{E}(w_i^t)^4 = \sum_{l=1}^{4} \binom{4}{l} \mathbb{E}\big[(w_i^t)^{4-l} \mathbb{E}_t[(\Delta w_i^t)^l]\big]$$

$$= 4\mathbb{E}\big[(w_i^t)^3 \mathbb{E}_t[\Delta w_i^t]\big] + 6\mathbb{E}\big[(w_i^t)^2 \mathbb{E}_t[(\Delta w_i^t)^2]\big] + 4\mathbb{E}\big[|w_i^t| \mathbb{E}_t[|\Delta w_i^t|^3]\big] + \mathbb{E}\big[\mathbb{E}_t[|\Delta w_i^t|^4]\big].$$

We bound the four terms on the right-hand side. For $l = 1$, we have

$$\mathbb{E}_t[\Delta_i^t] = -\eta \mathbb{E}_t\Big[g_i^t + \frac{\lambda}{d}\psi(w_i^t)\Big] = -\frac{\eta}{d}\big[(\lambda + \sigma_\psi^2)\psi(w_i^t) - \kappa_\psi w_i^*\big]$$

Hence, by Young's inequality,

$$\frac{1}{d}\sum_{i=1}^{d}\big|4\mathbb{E}[(w_i^t)^3 \mathbb{E}_t[\Delta w_i^t]]\big| \le \frac{C}{d}\Big(1 + \sum_i \mathbb{E}_t[(w_i^{t+1})^4]\Big)$$

For all $l = 2, 3, 4$, by Lemma D.5.1 and Young's inequality

$$\mathbb{E}\big[(w_i^t)^{4-l} \mathbb{E}_t[(\Delta w_i^t)^l]\big] \le \frac{C'}{d}\Big(1 + \sum_i \mathbb{E}_t(w_i^t)^4\Big).$$

Thus

$$\sum_i \mathbb{E}(w_i^{t+1})^4 - \sum_i \mathbb{E}(w_i^t)^4 \le \frac{C}{d}\Big(1 + \sum_i \mathbb{E}(w_i^t)^4\Big)$$

This

$$\sum_i \mathbb{E}(w_i^t)^4 \le \Big(1 + \frac{C}{d}\Big)^t \sum_i \mathbb{E}(w_i^0)^4 + \frac{C}{d}\sum_{k=0}^{t-1}\Big(1 + \frac{C}{d}\Big)^4 \le e^{CT}\Big(\sum_i \mathbb{E}w_i^0 + 1\Big) \le Cd$$

since $\sum_i \mathbb{E}w_i^0 \le Cd$ by Assumption C.1. □

### D.5.2. BOUND FOR MACROSCOPIC STATE

We next show that the macroscopic order parameters remain uniformly bounded over the time.

*Lemma* D.6. Under the same assumption as in Theorem 4.2, for all $t \le d \times T$, the following inequality holds:

$$\max_{0 \le \tau \le d \times T}\big\{\mathbb{E}q_t^2 + \mathbb{E}_t m_t^2\big\} \le C$$

*Proof.* From Lemma D.5 and Cauchy-Schwarz inequality,

$$\mathbb{E}q_t^2 = \frac{1}{d^2}\mathbb{E}\Big(\sum_i (w_i^t)^2\Big)^2 \le \frac{1}{d^2}d\mathbb{E}\sum_i (w_i^t)^4 \le C.$$

Similarly, for $m_t$ we apply Cauchy–Schwarz to obtain

$$\mathbb{E}m_t^2 = \frac{1}{d^2}\mathbb{E}\Big(\sum_i w_i^* w_i^t\Big)^2 \le \frac{\|\boldsymbol{w}^*\|^2}{d^2}\mathbb{E}\sum_i (w_i^t)^2 \le \frac{\rho^2}{d}\big(d\sum_i (w_i^t)^4\big)^{1/2} \le \rho^2\sqrt{C},$$

which concludes the whole proof. □

## E. Local Stability Analysis of Fixed Point of ODE

In the subsequent analysis, if the Jacobian matrix of the ODEs has only negative eigenvalues, the fixed point is termed locally stable. Conversely, if the Jacobian matrix has both zero and negative eigenvalues, the fixed point is referred to as marginally stable.

### E.1. Local Stability Analysis of Input-Quantized Models

We analyze the dynamics when only the inputs are quantized. In this setting, the order-parameter ODEs read

$$\frac{dm}{d\tau} = \eta\big(\kappa_\psi - (\sigma_\psi^2 + \lambda)m\big),$$

$$\frac{dq}{d\tau} = 2\eta\big(\kappa_\psi m - (\sigma_\psi^2 + \lambda)q\big) + \eta^2\sigma_\psi^2\,\varepsilon_g(m, q),$$

where the generalization error is

$$\varepsilon_g(m, q) \;=\; 1 + \sigma_\psi^2 q - 2\kappa_\psi m + \sigma^2.$$

**Fixed point.** Fixed points satisfy $(dm/d\tau, dq/d\tau) = (0, 0)$, yielding

$$m^* = \frac{\rho\kappa_\psi}{\sigma_\psi^2 + \lambda}, \quad q^* = \frac{2\kappa_\psi^2 + \eta\sigma_\psi^2\Big((\rho + \sigma^2)(\sigma_\psi^2 + \lambda) - 2\kappa_\psi^2\Big)}{(\sigma_\psi^2 + \lambda)\Big(2(\sigma_\psi^2 + \lambda) - \eta\sigma_\psi^4\Big)}.$$

Consequently, the fixed-point generalization error is

$$\varepsilon_g^* = \rho + \sigma^2 + \sigma_\psi^2 q^* - 2\kappa_\psi m^*.$$

In the small learning–rate limit,

$$\varepsilon_g^* \longrightarrow \rho + \sigma^2 - \frac{\kappa_\psi^2(\sigma_\psi^2 + 2\lambda)}{(\sigma_\psi^2 + \lambda)^2}.$$

The unquantized linear-regression case is recovered by setting $\kappa_\psi = \sigma_\psi^2 = 1$.

**Local stability.** Linearizing around $(m^*, q^*)$ yields the following Jacobian

$$J = \begin{pmatrix} -\eta(\sigma_\psi^2 + \lambda) & 0 \\ 2\eta\kappa_\psi - 2\eta^2\sigma_\psi^2\kappa_\psi & -2\eta(\sigma_\psi^2 + \lambda) + \eta^2\sigma_\psi^4 \end{pmatrix}.$$

The eigenvalues are

$$\lambda_1 = -\eta(\sigma_\psi^2 + \lambda), \qquad \lambda_2 = -2\eta(\sigma_\psi^2 + \lambda) + \eta^2\sigma_\psi^4.$$

Thus, the fixed point is locally stable whenever both eigenvalues are negative, i.e.,

$$0 < \eta < \frac{2(\sigma_\psi^2 + \lambda)}{\sigma_\psi^4}.$$

Note that the same condition also guarantees the denominator of $q^*$ is positive, ensuring the fixed point is well-defined.

### E.2. Fixed-Point Analysis of Weight-Input Quantized Models

We consider the macroscopic dynamics for $\Psi(\tau)$ in Theorem 5.3. The macroscopic ODEs are

$$\frac{dm(\tau)}{d\tau} = -\eta\big((\sigma_\psi^2 + \lambda)m_\psi(\tau) - \kappa_\psi\rho\big) \tag{25}$$

$$\frac{dq(\tau)}{d\tau} = -2\eta\big((\sigma_\psi^2 + \lambda)r_\psi(\tau) - \kappa_\psi m(\tau)\big) + \eta^2\sigma_\psi^2\varepsilon_g(\tau), \tag{26}$$

where
$$\varepsilon_g(\tau) = \rho - 2\kappa_\psi m_\psi(\tau) + \sigma_\psi^2 q_\psi(\tau) + \sigma^2,$$

and initial condition $\Psi(0) = \bar{\Psi}$. In the following, we characterize the fixed points $(m^*, q^*)$ through the fixed-point equation in terms of $s(\tau) := \sqrt{q(\tau) - m(\tau)^2}$.

Substituting the definition of $r_\psi$ and simplifying, we obtain that the fixed points satisfy $(dm/d\tau, dq/d\tau) = (0, 0)$, which yields

$$m_\psi(m, s) = c, \quad \frac{2s\kappa_\psi \Delta}{c} S_w(m, s) = \eta \sigma_\psi^2 \varepsilon_g(m, s), \quad c := \frac{\rho \kappa_\psi}{(\sigma_\psi^2 + \lambda)}.$$

We prove the following existence and uniqueness.

*Lemma* E.1. Fix $s > 0$ and $c \in (-\omega, \omega)$. The equation

$$m_\psi(m, s) = -\omega + \Delta \sum_{k=1}^{L} \Phi\left(\frac{m - \theta_k}{s}\right)$$

admits a unique solution $m = \mathsf{m}(s) \in \mathbb{R}$. Moreover,

$$\lim_{m \to -\infty} m_\psi(m, s) = -\omega, \quad \lim_{m \to +\infty} m_\psi(m, s) = +\omega,$$

and $m \mapsto m_\psi(m, s)$ is strictly increasing and continuous.

*Proof.* Each term $m \mapsto \Phi((m - \theta_k)/s)$ is continuous and strictly increasing; thus the finite sum is continuous and strictly increasing. As $m \to -\infty$, all $\Phi((m - \theta_k)/s) \to 0$, hence $m_\psi \to -\omega$. As $m \to +\infty$, all terms tend to 1, hence $m_\psi \to -\omega + \Delta L = \omega$. The intermediate value theorem yields existence for all $c \in (-\omega, \omega)$, and strict monotonicity yields uniqueness. $\square$

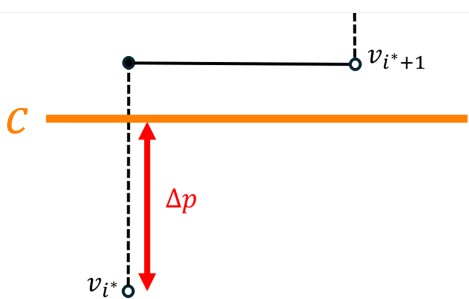

*Figure 7.* Conceptual illustration of the definition of index $i^*$ and the interpolation weight $p$.

Therefore, for each $s > 0$ and $|c| < \omega$ we can define the inverse map

$$\mathsf{m}(s) := \psi_s^{-1}(c) \quad \text{such that} \quad m_\psi(\mathsf{m}(s), s) = c.$$

Substituting $m = \mathsf{m}(s)$ gives a single-variable equation in $s$:

$$G_\eta(s) := \frac{2\Delta s}{\chi} S(\mathsf{m}(s), s) - \eta \varepsilon_g(\mathsf{m}(s), s) = 0, \quad \chi := \frac{\sigma_\psi^2}{\sigma_\psi^2 + \lambda}, \tag{27}$$

where

$$S(\mathsf{m}(s), s) := \sum_{k=1}^{L} \phi(z_k(m, s)), \quad z_k(m, s) := \frac{m - \theta_k}{s}$$

**Small $s$ Expansion**  However, it is difficult to further manipulate the nonlinear equation and carry out a fixed-point analysis. We therefore consider the small-$s$ expansion. From a learning perspective, the small-$s$ expansion corresponds to the limit in which the variance of the weight components orthogonal to the teacher is small. Intuitively, a smaller learning rate $\eta$ should reduce the variance of the orthogonal components, but the precise relationship between $s$ and $\eta$ is unclear. To clarify the meaning of the small-$s$ expansion, we first estimate the scaling (order) of $s$ with respect to $\eta$. To this end, we introduce the following notation to estimate the order of $s$ with respect to $\eta$.

**Definition E.2.** For any $c \in (-\omega, \omega)$ there exists a unique index

$$i^\star \in \{0, \ldots, L-1\}, \quad \text{such that} \quad v_{i^\star} < c < v_{i^\star+1},$$

and we denote the adjacent threshold by $\theta^* := \theta_{i^*+1}$. Define

$$p := \frac{c - v_{i^*}}{\Delta} \in (0, 1).$$

Equivalently, $c = (1-p)v_{i^*} + pv_{i^*+1} = -\omega + \Delta(i^* + p)$.

Figure 7 shows a conceptual diagram to illustrate the geometry of the quantities.

Since $i^\star = \lfloor (c+\omega)/\Delta \rfloor$, $p$ can be written as a function of $c$ as

$$p(c) = \frac{c+\omega}{\Delta} - \left\lfloor \frac{c+\omega}{\Delta} \right\rfloor,$$

i.e., the fractional part of $(c+\omega)/\Delta$. The motivation for this notation is the fixed-point equation

$$m_\psi(m, s) = \psi_s(m) = c, \tag{28}$$

from which, for fixed $s$, the value of $m$ is obtained by the de-quantization map

$$m = \psi_s^{-1}(c).$$

It follows from this equation that $m$ lies in a neighborhood of $\theta^\star$.

We quantify the behavior of the inverse $m = \mathsf{m}(s)$.

*Lemma E.3.* With $p \in (0, 1)$, the unique solution $\mathsf{m}(s)$ satisfies

$$\mathsf{m}(s) = \theta^* + s\Phi^{-1}(p) + \mathcal{O}\left(\frac{s^2}{\Delta}\phi\left(\frac{\Delta}{2s}\right)\right),$$

*Proof.* Let $z^* = (\mathsf{m}(s)-\theta^*)/s$. From Eq. (28) and $m_\psi(\mathsf{m}(s), s) = c = -\omega + \Delta(i^* + p)$,

$$p - \Phi(z^*) = \underbrace{\sum_{i=1}^{i^*} \Phi(z_i) - i^* + \Phi(z^*)}_{A(s)} + \underbrace{\sum_{i=i^*+2}^{L} \Phi(z_i)}_{B(s)},$$

By Lemma A.5,

$$0 \le i^* - A(s) \le i^* \frac{2s}{\Delta}\phi\left(\frac{\Delta}{2s}\right), \quad 0 \le B(s) \le (L - i^* - 1)\frac{2s}{\Delta}\phi\left(\frac{\Delta}{2s}\right)$$

hence

$$|p - \Phi(z^*)| \le C\frac{s}{\Delta}\phi\left(\frac{\Delta}{2s}\right).$$

Since $\Phi^{-1}$ is locally Lipschitz, there exists $a > 0$ with

$$|z^* - \Phi^{-1}(p)| \le \frac{1}{a}|\Phi(z^*) - p| \le C\frac{s}{\Delta}\phi\left(\frac{\Delta}{2s}\right)$$

Multiplying by $s$ and the definition $z^*$ yield the claim. □

*Lemma E.4.*

$$S(\mathsf{m}(s), s) = \phi(z^*) + \mathcal{O}\left(\phi\left(\frac{\Delta_w}{2s}\right)\right)$$

*Proof.* By definition, we have

$$S(s) = \phi(z^*) + \sum_{k \ne i^*+1} \phi(z_k).$$

From the definition, $|z_k| \ge \Delta/2s$ for $k \ne i^* + 1$, each tail term is bounded by $\phi(\Delta/2s)$. Summing over at most $L - 1$ indices gives the result. □

Next, we consider $q_\psi$.

*Lemma E.5.*

$$q_\psi(\mathsf{m}(s), s) = c^2 + \Delta_w^2 p(1 - p) + \mathcal{O}\left(s\phi\left(\frac{\Delta_w}{2s}\right)\right)$$

*Proof.* Similarly, we expand as follows:

$$q_\psi(m, s) = v_0^2 + \sum_{k=1}^{L} \underbrace{v_k^2 - v_{k-1}^2}_{\triangleq \Delta v_k^2} \Phi(z_k)$$

$$= v_0^2 + \sum_{k=1}^{i^*} \Delta v_k^2 \Phi(z_k) + \Delta v_{i^*+1}^2 \Phi(z^*) + \sum_{k=i^*+2}^{L} \Delta v_k^2 \Phi(z_k)$$

$$= v_0^2 + \sum_{k=1}^{i^*} \Delta v_k^2 (\Phi(z_k) - 1) + \sum_{k=1}^{i^*} \Delta v_k^2 + \Delta v_{i^*+1}^2 \Phi(z^*) + \sum_{k=i^*+2}^{L} \Delta v_k^2 \Phi(z_k)$$

$$= v_0^2 + \underbrace{\sum_{k=1}^{i^*} \Delta v_k^2}_{v_{i^*}^2} + \sum_{k=1}^{i^*} \Delta v_k^2 (\Phi(z_k) - 1) + \Delta v_{i^*+1}^2 \Phi(z^*) + \sum_{k=i^*+2}^{L} \Delta v_k^2 \Phi(z_k)$$

$$= v_{i^*}^2 + \underbrace{\sum_{k=1}^{i^*} \Delta v_k^2 (\Phi(z_k) - 1)}_{A(s)} + \Delta v_{i^*+1}^2 \Phi(z^*) + \underbrace{\sum_{k=i^*+2}^{L} \Delta v_k^2 \Phi(z_k)}_{B(s)}.$$

Since $|v_k^2 - v_{k-1}^2| \leq 2\omega\Delta$, we can bound $A(s), B(s)$ as follows:

$$|A(s)| \leq \sum_{k=1}^{i^*} 2\omega\Delta(1 - \Phi(z_k)) \leq 2\omega\Delta i^* \frac{2s}{\Delta}\phi\left(\frac{\Delta}{2s}\right)$$

$$|B(s)| \leq \sum_{k=i^*+2}^{L} 2\omega\Delta\Phi(z_k) \leq 2\omega\Delta(L - i^* - 1)\frac{2s}{\Delta}\phi\left(\frac{\Delta}{2s}\right).$$

Hence,

$$|A(s)| + |B(s)| \leq 2\omega(L-1)s\phi\left(\frac{\Delta}{2s}\right) = \mathcal{O}(se^{-\frac{\Delta^2}{8s^2}}).$$

Furthermore,

$$(v_{i^*+1}^2 - v_{i^*}^2)\Phi(z^*) = (v_{i^*+1}^2 - v_{i^*}^2)p + \underbrace{(v_{i^*+1}^2 - v_{i^*}^2)(\Phi(z^*) - p)}_{C(s)}.$$

$C(s)$ satisfies

$$C(s) = |(v_{i^*+1}^2 - v_{i^*}^2)(\Phi(z^*) - p)| \leq 4\omega(L-1)s\phi\left(\frac{\Delta}{2s}\right)$$

Since $v_{i^*+1} = v_{i^*} + \Delta$, it follows that

$$q_\psi(m, s) = v_{i^*}^2 + (v_{i^*+1}^2 - v_{i^*}^2)p + \mathcal{O}\left(se^{-\frac{\Delta^2}{8s^2}}\right)$$

$$= v_{i^*}^2 + (2v_{i^*}\Delta + \Delta^2)p + \mathcal{O}\left(se^{-\frac{\Delta^2}{8s^2}}\right)$$

$$= \underbrace{(v_{i^*} + p\Delta)^2}_{c^2} + \Delta^2 p(1-p) + \mathcal{O}\left(se^{-\frac{\Delta^2}{8s^2}}\right)$$

$$= c^2 + \Delta^2 p(1-p) + \mathcal{O}\left(se^{-\frac{\Delta^2}{8s^2}}\right)$$

$\square$

*Proposition* E.6. With $c \leq \omega$,

$$\varepsilon_g = \rho + \sigma^2 - 2\kappa_\psi c + \sigma_\psi^2 c^2 + \sigma_\psi^2 \Delta^2 p(1-p) + \mathcal{O}\left(s\phi\left(\frac{\Delta}{2s}\right)\right).$$

*Proof.* From the fixed-point equation $m_\psi(\mathsf{m}(s), s) = c$, we have

$$\varepsilon_g(\mathsf{m}(s), s) = (1 - c)^2 + (q_\psi - c^2) + \sigma^2 = (1 - c^2) + \Delta^2 p(1 - p) + \mathcal{O}\left(s\phi\left(\frac{\Delta}{2s}\right)\right)$$

$\square$

*Proposition E.7.* If $|c| < \omega$, the equation $G_\eta(s) = 0$ admits a solution with

$$s(\eta) = \Theta_\eta(\eta) \quad p \in (0, 1).$$

$$s(\eta) = \Theta_\eta\left(\frac{1}{\sqrt{\log(1/\eta)}}\right), \quad p \in \{0, 1\}$$

Specifically, when $p \in (0, 1)$,

$$s(\eta) = \frac{\chi}{2\Delta_\omega \phi(z^*)} \eta + o(\eta),$$

where

$$\varepsilon_g^{(0)} := 1 + \sigma^2 - 2\kappa_\psi c + \sigma_\psi^2 c^2 + \sigma_\psi^2 \Delta_w^2 p(1 - p).$$

*Proof.* Substituting these into the fixed-point equation gives

$$G_\eta(s) = \frac{2s\Delta}{c}\left(\phi(z^*) + \mathcal{O}\left(\phi\left(\frac{\Delta}{2s}\right)\right)\right) - \eta\left(\underbrace{(1 - c^2) + \Delta^2 p(1 - p)}_{\varepsilon_g^{(0)}} + \mathcal{O}\left(s\phi\left(\frac{\Delta}{2s}\right)\right)\right) = 0$$

Therefore,

$$\frac{2s\Delta}{c}\phi(z^*) - \eta\varepsilon_g^{(0)} = \mathcal{O}\left(e^{-\frac{\Delta^2}{8s^2}}\right)$$

Hence,

$$\lim_{\eta \to 0} \frac{s(\eta)}{\eta} = \frac{\varepsilon_g^{(0)}}{2(1 + \lambda)\Delta\phi(z^*)}.$$

When $p \in \{0, 1\}$, Lemma E.4 yields $S(\mathsf{m}(s), s) = \Theta(\phi(\Delta/(2s)))$. Eq. (27) becomes

$$s\phi\left(\frac{\Delta}{2s}\right) \asymp \eta$$

and substituting $\phi(t) = (2\pi)^{-1/2}e^{-t^2/2}$ with $t = \Delta/2s$ gives

$$e^{-\frac{\Delta^2}{8s^2}} \asymp \eta \implies s(\eta) = \frac{\Delta}{2\sqrt{2\log(1/\eta)}}(1 + o_\eta(1)).$$

The asserted order for $s(\eta)\phi(\Delta/2s(\eta))$ then follows by direct substitution. $\square$

In the above section, we can estimate the order of parameter $s$. Here, we evaluate the generalization error at the fixed point for small $\eta$.

**Theorem E.8.** *In the small-learning-rate limit, the generalization error $\varepsilon_g^*$ satisfies*

$$\varepsilon_g^* = \begin{cases} \varepsilon_g^{(0)} + \sigma_\psi^2 \Delta^2 p(1 - p) + o(\eta), & |c| < \omega, p \in (0, 1), \\ \varepsilon_g^{(0)} + o(1/\sqrt{\log(1/\eta)}), & |c| < \omega, p \in \{0, 1\}, \\ \rho + \sigma^2 - 2\kappa_\psi\omega + \sigma_\psi^2\omega^2 + o(\eta), & |c| \geq \omega, \end{cases}$$

*where $\varepsilon_g^{(0)} = \rho + \sigma^2 - 2\kappa_\psi c + \sigma_\psi^2\omega^2$ denotes the generalization error of the input-only quantized model in the small-learning-rate limit.*

*Proof.* We treat the three regimes separately.

*Case 1:* $|c| < \omega$ *and* $p \in (0, 1)$. By Proposition E.7, the fixed-point solution satisfies $s(\eta) = \Theta(\eta)$. Substituting $(m(s(\eta)), s(\eta))$ into Proposition E.7 yields

$$\varepsilon_g^* = \varepsilon_g^{(0)} + \mathcal{O}_{s(\eta)}\left( s(\eta)\phi\left( \frac{\Delta}{2s(\eta)} \right) \right).$$

Since $\phi(\Delta/2s)$ is exponentially small in $1/\eta^2$, the remainder is $o_\eta(\eta)$.

*Case 2:* $|c| < \omega$ *and* $p \in \{0, 1\}$. Proposition E.7 yields

$$s(\eta)\phi\left( \frac{\Delta}{2s(\eta)} \right) = \Theta\left( \frac{\eta}{\sqrt{\log(1/\eta)}} \right),$$

which implies that

$$\varepsilon_g^* = \varepsilon_g^{(0)} + \mathcal{O}_{s(\eta)}\left( s(\eta)\phi\left( \frac{\Delta}{2s(\eta)} \right) \right) = \varepsilon_g^{(0)} + o_\eta\left( \frac{1}{\sqrt{\log(1/\eta)}} \right),$$

Since $\eta/\sqrt{\log(1/\eta)} = o_\eta(1/\sqrt{\log(1/\eta)})$, we obtain the second claim.

*Case 3:* $|c| \geq \omega$. Since $m_\psi(m, s) \in [-\omega, \omega]$ for all $m$, $s$, the interior constraint $m_\psi = c$ is infeasible. In the small-$\eta$ limit, the first stationarity condition is replaced by its constrained counterpart: the unique minimizer of the affine function $\mu \mapsto (\sigma_\psi^2 + \lambda)\mu - \rho\kappa_\psi$ over $\mu \in [-\omega, \omega]$ is attained at the boundary $\mu = \text{sign}(c)\omega$. Since $c \geq 0$ under the assumption, the relevant boundary is $+\omega$. Therefore any small-$\eta$ fixed point satisfies

$$m_\psi(m, s) = \omega + o_\eta(1), \quad q_\psi(m, s) = \omega^2 + o_\eta(1), \quad s \to +0,$$

where the last convergence follows from the same tail estimates used earlier, since all terms except the final one are negligible while the last decays as $\phi(\Delta/2s)$. Substituting into $\varepsilon_g$ yields

$$\varepsilon_g^* = \rho + \sigma^2 - 2\kappa_\psi\omega + \sigma_\psi^2\omega^2 + r(\eta),$$

with $|r(\eta)| \leq C(|\omega - m_\psi| + |\omega^2 - q_\psi|)$. Proposition E.7 implies that

$$s(\eta)\phi\left( \frac{\Delta}{2s(\eta)} \right) = \Theta_\eta\left( \frac{\eta}{\sqrt{\log(1/\eta)}} \right),$$

where $|\omega - m_\psi| + |\omega^2 - q_\psi| = o_\eta(\eta)$. Therefore $r(\eta) = o_\eta(\eta)$, which proves the third claim. $\square$

## F. Relation to Layer-wise PTQ

The analysis of the linear regression model is closely related to layer-wise post-training quantization (PTQ), a recently emerging approach for LLM quantization that has gained widespread adoption after pre-training. In layer-wise PTQ (Lin et al., 2024; Achiam et al., 2023; Zhao et al., 2025), each linear layer of the LLM is quantized by solving the following optimization problem:

$$\min_{\hat{W} \in \mathbb{R}^{m \times d}} \left[ \frac{1}{2} \sum_{\mu=1}^n \left\| W\boldsymbol{x}_\mu - \hat{W}\boldsymbol{x}_\mu \right\|^2 + \frac{\lambda}{2}\|\hat{W}\|_F^2 \right],$$

where $W$ denotes the pre-quantization weights, and $\hat{W}$ denotes the quantized weights. When we set $m = 1$ and identify vectors so that $\boldsymbol{w} := \psi(\boldsymbol{w}; b, \omega)$ and $\boldsymbol{w}^* = \psi(\boldsymbol{w}; b, \omega)$ with $\boldsymbol{w}^* = \boldsymbol{w}$, the formulation coincides up to the scaling factor $1/\sqrt{d}$. This correspondence, therefore, provides insight into the subproblems that arise in layer-wise PTQ. Moreover, recent work argues that the following objective should be solved when applying layer-wise PTQ (Arai & Ichikawa, 2025):

$$\min_{\hat{W} \in \mathbb{R}^{m \times d}} \left[ \frac{1}{2} \sum_{\mu=1}^n \left\| W\boldsymbol{x}_\mu - \hat{W}\hat{\boldsymbol{x}}_\mu \right\|^2 + \frac{\lambda}{2}\|\hat{W}\|_F^2 \right]$$

where $\hat{\boldsymbol{x}}$ denotes the activation generated by the quantized weights of the preceding layers, i.e., the input to the current layer. The quantization error is scalar in our setting and can be partially represented by the noise term $\xi_\mu$. Building on this analysis, assessing the performance of layer-wise PTQ represents an interesting direction for future work.

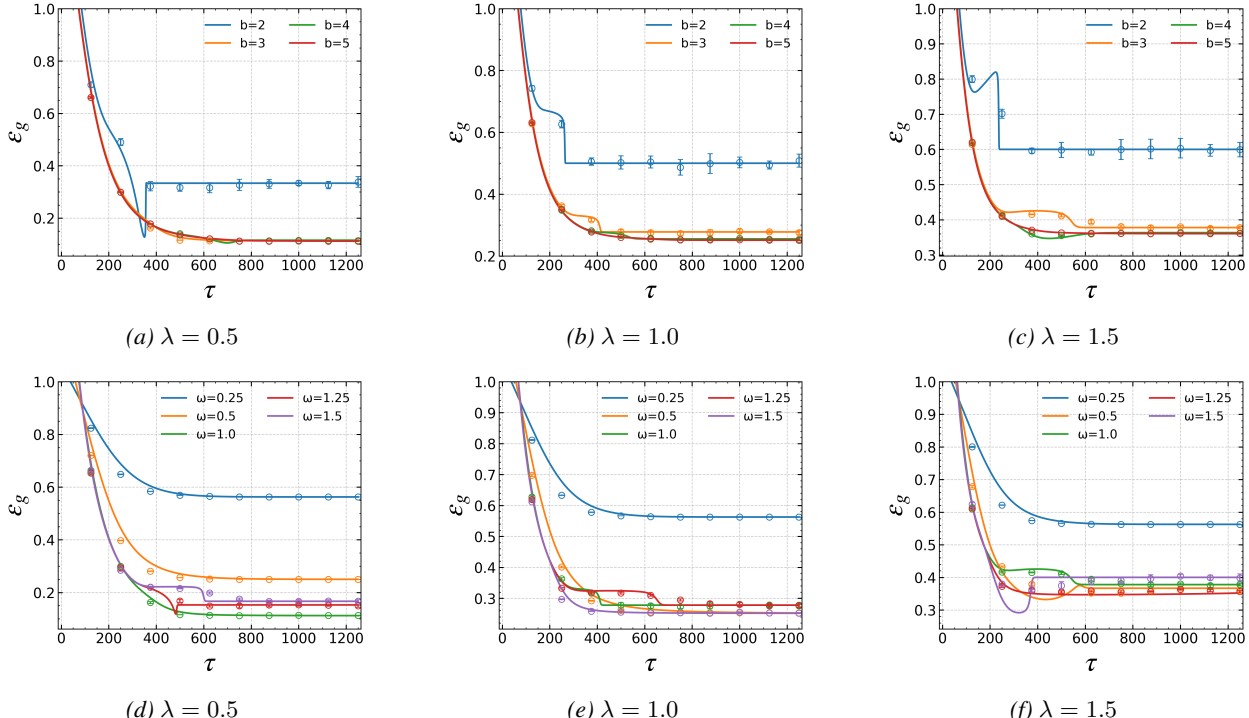

*Figure 8.* Each panel shows the generalization error $\varepsilon_g$ as a function of training time $\tau$. *Top row:* effect of bit width $b \in \{2, 3, 4, 5\}$ at fixed quantization range $\omega = 1.0$. *Bottom row:* effect of quantization range $\omega$ at fixed bit width $b = 3$. Columns are ordered left to right by $\lambda$. Solid curves show the ODE predictions; circular markers denote empirical averages from the STE simulation in dimension $d = 900$ averaged over five runs.

## G. Additional Experiments

### G.1. Dependence on the Ridge Regularization Parameter

We study how the ridge coefficient $\lambda$ influences the learning dynamics of a weight-only quantized linear model. We report the generalization error $\varepsilon_g(\tau)$ as a function of training time $\tau$. In the numerical STE simulation, the input dimension is fixed at $d = 900$, and each curve represents an average over five independent runs. We compare the empirical STE trajectories with the deterministic ODE predictions under the isotropy assumption. The ablation study is organized into two complementary parts, as shown in Figure 8. In the *top row*, the quantization range is fixed at $\omega = 1.0$ while the bit width varies as $b \in \{2, 3, 4, 5\}$; in the *bottom row*, we fix $b = 3$ and vary the quantization range $\omega \in \{0.25, 0.50, 1.00, 1.25, 1.50\}$. From left to right, the columns correspond to $\lambda \in \{0.5, 1.0, 1.5\}$. Across all panels, the ODE predictions closely follow the empirical STE trajectories.

**Effect of Bit Width on Fixed** $\omega = 1.0$. Increasing the bit width consistently reduces both transient and steady-state errors. The improvement is most notable when increasing from $b = 2$ to $b = 3$, while gains beyond $b = 4$ are marginal: the $b = 4$ and $b = 5$ trajectories exhibit similar behavior in $\lambda \in \{0.5, 1.0\}$ and remain very close even at $\lambda = 1.5$.

**Effect of Quantization Range at Fixed** $b = 3$. The choice of $\omega$ influences the achievable generalization error. Small ranges, $\omega \in \{0.25, 0.50\}$ saturate early and lead to significantly higher plateaus. Moderate to large ranges, $\omega \in \{1.0, 1.25, 1.5\}$, yield much lower errors, with $\omega \approx 1.0 \sim 1.25$ performing best at $\lambda \in \{0.5, 1.0\}$. Thus, beyond the bit budget, the quantization range is a key parameter for controlling quantization performance.

**Practical Implication: Early Stopping.** At $\lambda = 1.5$, the intermediate minima of $\varepsilon_g(\tau)$ are significantly lower than the eventual steady-state values. This indicates that *early stopping*, which involves selecting the iterate at the transient minimum, results in better generalization compared to training until convergence. For smaller $\lambda$, the trajectories are nearly monotonic, and the gap between the transient and final errors diminishes, rendering early stopping less critical.

*Table 1.* Relative error between the order parameters predicted under the isotropy closure (Section 5) and finite-dimensional STE simulation, with the residual skewness of the orthogonal weight component. Weight-only quantization, $\omega = 1$, evaluated at $\tau = 200$ and averaged over five runs.

| $b$ | $d$ | $m_\psi$ err | $q_\psi$ err | $r_\psi$ err | Skewness |
|---|---|---|---|---|---|
| 4 | 500 | 0.33% | 0.70% | 0.35% | $-0.02 \pm 0.14$ |
| 4 | 1000 | 0.26% | 0.51% | 0.25% | $-0.05 \pm 0.12$ |
| 4 | 2000 | 0.19% | 0.39% | 0.19% | $+0.05 \pm 0.08$ |
| 3 | 500 | 0.38% | 0.68% | 0.42% | $-0.05 \pm 0.20$ |
| 3 | 1000 | 0.27% | 0.48% | 0.30% | $+0.02 \pm 0.30$ |
| 3 | 2000 | 0.40% | 0.72% | 0.42% | $-0.04 \pm 0.19$ |

*Table 2.* Generalization error $\varepsilon_g$ at $\tau = 200$ for the all-ones teacher $\boldsymbol{w}^* = \mathbf{1}_d$ and an i.i.d. Gaussian teacher, with $d = 1000$ and five runs. Top: weight-only quantization ($\omega = 1$), indexed by the weight bit width $b$. Bottom: joint weight–input quantization ($\eta = 0.05$), indexed by $(b_w, b_x)$.

| | $\varepsilon_g \, (\boldsymbol{w}^* = \mathbf{1})$ | $\varepsilon_g \, (\text{Gaussian } \boldsymbol{w}^*)$ |
|---|---|---|
| *Weight-only quantization* | | |
| $b = 3$ | $0.273 \pm 0.006$ | $0.295 \pm 0.003$ |
| $b = 4$ | $0.255 \pm 0.003$ | $0.282 \pm 0.002$ |
| *Joint weight–input quantization* | | |
| $(b_w, b_x) = (3, 3)$ | $0.504 \pm 0.003$ | $0.512 \pm 0.008$ |
| $(b_w, b_x) = (3, 4)$ | $0.502 \pm 0.005$ | $0.514 \pm 0.008$ |
| $(b_w, b_x) = (4, 4)$ | $0.494 \pm 0.003$ | $0.509 \pm 0.003$ |

## G.2. Accuracy of the Isotropy Closure

The closed macroscopic ODE in Theorem 5.3 relies on the isotropy closure of Assumption 5.1, which is exact for affine single-coordinate drifts but only approximate once the hard quantizer introduces a threshold nonlinearity. To assess its adequacy directly, we compare the nonlinear order parameters $m_\psi$, $q_\psi$, and $r_\psi$ predicted by the closed-form expressions in Section 5, which use the closure, against finite-dimensional STE simulations. We use the weight-only quantized model with $\omega = 1$, evaluate at $\tau = 200$, and average over five independent runs. Table 1 reports the relative error of each order parameter, together with the sample skewness of the orthogonal weight component, which the closure predicts to be zero. Across all tested bit widths and dimensions, every relative error remains below $1\%$ and decreases as $d$ grows, while the residual skewness is statistically indistinguishable from zero. These diagnostics confirm that the isotropy closure accurately reproduces the quantities that enter both the ODE and the generalization error in the regimes studied.

## G.3. Robustness to the Teacher Distribution

Several of our illustrative experiments use the all-ones teacher $\boldsymbol{w}^* = \mathbf{1}_d$, which is aligned with the quantization grid and therefore not without loss of generality once component-wise quantization breaks the rotational symmetry. The microscopic SDE/PDE limit of Section 4 and Theorem 5.3 nonetheless hold for any i.i.d. teacher $w_i^* \sim p_{w^*}$ with $\|\boldsymbol{w}^*\|^2 = \rho d$. To verify that the conclusions are not an artifact of the aligned teacher, we repeat the dynamics with an i.i.d. Gaussian teacher and compare the generalization error against the all-ones case under otherwise identical settings ($d = 1000$, $\tau = 200$, five runs). Table 2 reports both the weight-only ($\omega = 1$) and the joint weight–input ($\eta = 0.05$) settings. The Gaussian teacher yields slightly larger errors, as expected from its less grid-aligned mass distribution, but the qualitative ordering across bit widths is preserved, indicating that the reported phenomena are robust to the choice of teacher.

## G.4. Beyond the Linear Model: Nonlinear Networks and Real Data

Our theory is developed for a linear Gaussian teacher–student model, and a natural question is whether the predicted effect of the quantization range on training stability survives in nonlinear architectures and on real data. To probe this, we train a two-layer ReLU network $\hat{y} = \frac{1}{\sqrt{K}} \sum_{k=1}^{K} a_k \, \text{ReLU}(\boldsymbol{w}_k^\top Q_{b_x, \omega_x}(\boldsymbol{x}) / \sqrt{d})$ with hidden width $K = 32$, using online SGD on the squared loss with input-only quantization ($b_x = 2$) and averaging over eight runs. We estimate the maximal stable

*Table 3.* Maximal stable learning rate $\eta_{\max}$ for a two-layer ReLU network ($K = 32$, input-only quantization with $b_x = 2$, eight runs) on the diabetes dataset ($d = 10$), as a function of the input quantization range $\omega_x$. The bottom row reports the relative change with respect to the full-precision baseline.

|  | Unquantized | $\omega_x = 1.0$ | $\omega_x = 1.4$ | $\omega_x = 2.0$ | $\omega_x = 4.5$ |
|---|---|---|---|---|---|
| $\eta_{\max}$ | $0.016 \pm .001$ | $\mathbf{0}.021 \pm .002$ | $0.017 \pm .001$ | $0.013 \pm .001$ | $0.010 \pm .001$ |
| vs. full-precision | — | $+35\%$ | $+8\%$ | $-16\%$ | $-35\%$ |

*Table 4.* Maximal stable learning rate $\eta_{\max}$ for a two-layer ReLU network ($K = 32$, input-only quantization with $b_x = 2$, eight runs) on three real-world regression benchmarks, as a function of the input quantization range $\omega_x$. Each dataset is annotated with its input dimension and sample size $(d, n)$.

| Dataset $(d, n)$ | Unquantized | $\omega_x = 1.0$ | $\omega_x = 1.4$ | $\omega_x = 2.0$ | $\omega_x = 4.5$ |
|---|---|---|---|---|---|
| California Housing $(8, 20k)$ | $0.37 \pm .17$ | $\mathbf{2}.81 \pm .24$ | $2.13 \pm .22$ | $1.75 \pm .25$ | $2.25 \pm .35$ |
| Superconductivity $(81, 21k)$ | $2.38 \pm .22$ | $\mathbf{3}.00 \pm .00$ | $2.88 \pm .22$ | $2.38 \pm .22$ | $2.75 \pm .25$ |
| CPU Activity $(21, 8k)$ | $0.48 \pm .13$ | $\mathbf{2}.00 \pm .00$ | $2.00 \pm .00$ | $1.69 \pm .24$ | $1.50 \pm .25$ |

learning rate $\eta_{\max}$, i.e., the largest learning rate for which training does not diverge, as a function of the input quantization range $\omega_x$. Table 3 reports the diabetes dataset, and Table 4 reports three further regression benchmarks. Across all datasets, a moderate range ($\omega_x = 1.0$) consistently enlarges the stable region relative to the unquantized baseline, whereas a large range ($\omega_x = 4.5$) erodes or reverses this gain. This reproduces the widen-then-shrink stability transition predicted by the linear theory (Figure 5) in a nonlinear model on realistic tasks, indicating that the implicit-regularization effect of input quantization persists beyond the stylized setting.

Finally, we examine a classification setting to delineate the scope of the stability mechanism. We train a logistic model $\hat{p} = \sigma(\boldsymbol{w}^\top Q_{b_x, \omega_x}(\boldsymbol{x})/\sqrt{d})$ on the breast-cancer dataset ($d = 30$) with online SGD on the binary cross-entropy loss and input-only quantization ($b_x = 2$), averaging over eight runs. Here training remained stable up to $\eta = 128$ for all ranges, so the sharp $\eta_{\max}$ shift observed for the squared loss does not appear; this is consistent with our analysis, since the sigmoid saturation bounds the gradient and suppresses the divergence mechanism that underlies the boundary $\eta_{\max} \propto 1/\sigma_\psi^4$ in regression. The quantization range nevertheless remains a first-order hyperparameter for the final quality: as shown in Table 5, the test accuracy degrades by more than 20 percentage points as $\omega_x$ increases from 1.0 to 4.5.

*Table 5.* Test accuracy of a logistic model with input-only quantization ($b_x = 2$, eight runs) on the breast-cancer dataset ($d = 30$), as a function of the input quantization range $\omega_x$.

|  | Unquantized | $\omega_x = 1.0$ | $\omega_x = 1.4$ | $\omega_x = 2.0$ | $\omega_x = 4.5$ |
|---|---|---|---|---|---|
| Accuracy | $95.5\%$ | $92.3\%$ | $90.8\%$ | $83.7\%$ | $74.3\%$ |

